# Mapping of QTLs Associated with Yield and Yield Related Traits in Durum Wheat (*Triticum durum* Desf.) Under Irrigated and Drought Conditions

**DOI:** 10.3390/ijms21072372

**Published:** 2020-03-30

**Authors:** Mian Abdur Rehman Arif, Fauzia Attaria, Sajid Shokat, Saba Akram, Muhammad Qandeel Waheed, Anjuman Arif, Andreas Börner

**Affiliations:** 1Nuclear Institute for Agriculture and Biology, Jhang Road, Faisalabad, 38000, Pakistan; afsmta@gmail.com (F.A.); sajid_agrarian@yahoo.com (S.S.); sabaakram3463@yahoo.com (S.A.); qandeel.nibgian@gmail.com (M.Q.W.); arifanjuman@gmail.com (A.A.); 2Department of Plant and Environmental Sciences, University of Copenhagen, Højbakkegård Allé 13, DK-2630 Taastrup, Denmark; 3Leibniz Institute of Plant Genetics and Crop Plant Research, Corrensstr. 3, Seeland OT, 06466 Gatersleben, Germany; boerner@ipk-gatersleben.de

**Keywords:** *Triticum durum*, QTL mapping, drought stress, yield, stress indices, marker assisted selection

## Abstract

Global durum wheat consumption (*Triticum durum* Desf.) is ahead of its production. One reason for this is abiotic stress, e.g., drought. Breeding for resistance to drought is complicated by the lack of fast, reproducible screening techniques and the inability to routinely create defined and repeatable water stress conditions. Here, we report the first analysis of dissection of yield and yield-related traits in durum wheat in Pakistan, seeking to elucidate the genetic components of yield and agronomic traits. Analysis of several traits revealed a total of 221 (160 with logarithm of odds (LOD) > 2 ≤ 3 and 61 with LOD > 3) quantitative trait loci (QTLs) distributed on all fourteen durum wheat chromosomes, of which 109 (78 with LOD > 2 ≤ 3 and 31 with LOD > 3) were observed in 2016-17 (S1) and 112 (82 with LOD > 2 ≤ 3 and 30 with LOD > 3) were observed in 2017-18 (S2). Allelic profiles of yield QTLs on chromosome 2A and 7B indicate that allele A of *Xgwm895* and allele B of *Xbarc276* can enhance the Yd up to 6.16% in control and 5.27% under drought. Moreover, if combined, a yield gain of up to 11% would be possible.

## 1. Introduction

Wheat has been the world’s most important cereal from 10,000 years ago since the dawn of agriculture [1] up to the present day. With a production of ~620 million tons annually worldwide, wheat provides about 1/5th of the calories consumed by humans [2]. The current annual wheat yield increase is below 1%, but it should be around 1.6% in order to meet the wheat demand (which is anticipated to rise by 60% with the world population, which will be 9 billion by 2050 [3].

Durum wheat (*Triticum durum or Triticum turgidum subsp. durum*), which is also referred to as pasta or macaroni wheat, is the hardest of all wheats. Its density, combined with its high protein content and gluten strength, make durum the wheat of choice for producing quality products including bread, couscous, frekeh, bulgur, and most importantly pasta [4]. A total of 9.3 and 10.5 million tons of pasta was produced in 2001 and 2003, respectively. By 2013, its production reached 13.5 million tons [5], providing an insight into the growing global demand for durum wheat.

Durum wheat global consumption is ahead of its production. *T. durum* accounts for around 6% of total wheat production (37.7 MT in 2013; International Grain Council, October 2014) [6], occupying approximately 20 million hectares worldwide. High yielding cultivars endowed with drought tolerance and disease resistance, in addition to their high commercial and technological value, are therefore highly desirable. Durum wheat is primarily grown under rain-fed conditions where the frequent drought combined with heat stress is the major aspect of grain yield reduction [7,8,9]. Drought is one of the most severe stresses and is the main cause of significant losses in growth and productivity of crop plants [10]. About 45% of the 120 million hectares of land allocated in developing countries to wheat production is prone to drought [11]. Crop yield is seriously affected by water deficit in many growing areas of the world. It has been assessed that as much as 50% of the wheat cultivation area is regularly affected by drought [12].

In response to water deficit, morphological and biochemical changes are witnessed in plants which follow the functional damage and loss of important plant parts as the shortfall increases [13]. Various tolerance mechanisms (pigment content and stability and high relative water content) have been ascribed to the physiological changes in plants to water scarcity [14]. There are various ways in which drought-tolerant wheat species can be differentiated. These include growth response in water-limited conditions, alteration in water relations in tissues subjected to water stress, amassing of various ions, and early flowering [15]. As our comprehension of the physiological processes by which plants respond to water scarcity increases, we will be able to devise screening methods to select drought-tolerant accessions for further usage in breeding systems [16]. 

Generally, all yield components are disturbed under severe drought stress, most notably including fertile ears per unit area (reduction ≤ 60%), seeds per spike (reduction ≤ 60%), harvest index; and nitrogen harvest index (reduction ≤ 24%), whereas lower grain weight is witnessed under mild stress [17]. Different cultivars differ in their responses to drought. Several factors (CO_2_ exchange rate, stomatal conductance, relative water contents and osmotic changes) regulate the fate of drought resistance in a genotype and natural variation in these factors can be used as selection criteria to breed drought-tolerant cultivars [18]. Since the 1970s, the attained genetic gains in durum yield have been because of well-adjusted progress in fertility through distribution of relatively higher assimilates to the ear and the growing tillers which increased total biomass, whereas harvest index remained the same [19], probably because the ratio of improvement of grain yield to biological yield remained the same. Water usage of plants and minimization of the deleterious effects under water-scarce conditions can be modified by altering plant height, exposed leaf area and leaf index [20]. Moreover, varieties enriched with increased leaf turgidity and relative water contents in water-deficient environments have better yields in drought [21,22]. 

Several factors are responsible for complications in breeding drought-tolerant cultivars, such as fast repeatable screening techniques and the incapacity to create drought conditions in a repeatable manner to screen large amounts of germplasm [23]. Breeders look for yield performance under drought stress. However, there are other factors that lead to poor yield in drought. To nullify this, drought indices have been devised based on plants performance under normal and drought conditions [24]. The drought resistance or susceptibility of a given genotype forms the basis of these indices [25]. Resistance to drought of a given genotype can be defined as the relative yield of that genotype in comparison to other genotypes that have been grown under the exact same stress conditions [26]. 

Different mapping populations have been made to investigate both biotic (disease resistance) and abiotic (drought, salinity) stresses [27,28], and many QTLs have been identified for yield and related traits in field trails. For example, two QTLs for test weight on chromosomes 7A and 6B (explaining 30% variation) and five QTLs of thousand kernel weight (Tkw) (explaining 32% of total, including 25% genetic, variation) have been revealed [17]. Other investigations in durum wheat have reported QTLs for yield and its components and physiological developmental [29,30]. Although a plethora of information regarding genetic maps, populations and molecular markers exists, the transformation of this information into the breeding of drought-tolerant cultivars cannot yet be compared. This anomaly can be attributed to three reasons. The first is that yield, being a quantitative trait, has vague genetic architecture. The second is that the yield-linked loci are subject to a strong environmental influence. The third is the poor phenotyping of the concerned traits. Established QTLs in one environment often disappear in a different environment, and thus QTLs have been well-defined as constitutive versus adaptive. Nevertheless, robust QTLs of yield traits across 16 different environments on chromosomes 2B and 3B have been reported [29].

Here, we report the analysis of a bi-parental mapping population seeking to elucidate the genetic components of yield and agronomic traits under irrigated and drought conditions. Moreover, certain physiological traits were analyzed to check their response towards yield in both conditions. To add to this, various stress indices were calculated and their QTLs were sought. Thus, the current study was undertaken to achieve the following objectives: (1) to screen a panel of 114 recombinant inbred lines (RILs) of durum wheat for yield and yield-related traits under drought and irrigated conditions at the Nuclear Institute for Agriculture and Biology (NIAB), Pakistan; (2) to measure physiological traits and various stress indices of the population under irrigated and drought conditions; and (3) to uncover the potential QTLs of measured traits and their comparison.

## 2. Results

### 2.1. Phenotypic Variation

A three-way ANOVA revealed that treatment, i.e., drought, significantly reduced all the traits (Table 1). Moreover, germination (Gr), transpiration rate (Tr), stomatal conductance (Sc), photosynthetic rate (Phr), yield (Yd), harvest index (HI), and thousand kernel weight (Tkw) differed significantly at the genotype level. Impact of season was significant for Gr, Sc, Phr and Sl, where an increase for Gr was observed in S2, while Sc Phr and Sl were significantly higher in S1. Furthermore, genotype x treatment interaction was significant for Tr, Sc, Ph and HI. Season x treatment had a significant effect on Gr, Sc, Phr, Wue, (heading time) Hd and Sl. Finally, genotype x season x treatment were significant for Sc and Yd. 

### 2.2. Agronomic Traits

Mean ± SD along with range of the traits are provided in Table 1, whereas box plots are given in Appendix A. Agronomic traits included Gr, Hd and Ph. S1 (2016-17) and S2 (2017-18) indicate different growing seasons. Mean germination under control in S1 reduced from 81.86 ± 9.63 (Gr_S1) to 72.55 ± 9.96 (GrD_S1), with relative germination being 88.2 4 ± 11.78 (GrD_S1). Similarly, mean germination under control in S2 decreased from 96.00 ± 3.72 (Gr_S2) to 91.24 ± 5.56 (GrD_S2) in drought, with relative germination being 95.11 ± 5.30. Heading time in control in S1 decreased from 100.47 ± 11.79 (Hd_S1) to 93.94 ± 12.41 9 (HdD_S1) in drought. The same decreased from 100.47 ± 11.79 (Hd_S1) to 92.32 ± 6.79 (HdD_S2) under drought in S2. The relative heading times in S1 and S2 were 94.30±7.38 and 94.18 ± 2.70, respectively. Reduction in plant height was also observed in drought as compared to control, as it was decreased from 114.78 ± 23.82 (Ph_S1) in control to 80.24 ± 12.45 (PhD_S1) in drought in S1. Likewise, it was decreased from 111.36 ± 26.31 (Ph_S2) to 83.65 ± 14.14 (PhD_S2) in drought in S2. (not mark in total)

### 2.3. Physiological Traits

These included Tr, Phr, Sc and Wue, which were higher in control than their corresponding values in drought in both seasons. In S1, Tr dropped form 37.38 ± 6.86 (Tr_S1) to 15.56 ± 4.65 (TrD_S1) under drought. On the same ground, in S2 it dropped from 34.17 ± 6.51 (Tr_S2) to 15.06 ± 1.90 (TrD_S2). However, RTr_S1 (42.98 ± 15.98) was lower than RTr_S2 (45.43 ± 9.78). Phr reduced from 3.97 ± 1.24 (Phr_S1) to 2.22 ± 0.44 (PhrD_S1) in drought. Likewise, a reduction from 3.49 ± 1.25 (Phr_S2) to 1.72 ± 0.67 (PhD_S2) in drought was observed in S2. RPhr_S1 (58.96 ± 15.51) was higher than RPhr_S2 (50.94 ± 15.76). In control, Wue in control treatment in S1 was 11.89 ± 4.03 (Wue_S1), which increased to 16.04 ± 7.93 in drought, whereas it was 11.14 ± 6.30 (Wue_S2) in control treatment in S2, which increased to 11.43 ± 5.14 (WueD_S2) in drought. This also raised the mean RWue_S1 (144.41 ± 87.62) and RWue_S2 (113.98 ± 41.44). 

### 2.4. Spike-Related Traits

These included Sl, seeds per spike (Sps), spike weight (Sw) and Tkw. As per spike-related traits, the values of Sl, Sw and Tkw were higher in S1, whereas Sps was higher in S2. In drought, spike length was dropped to 7.13 ± 1.11 (SlD_S1) from 8.25 ± 0.96 (Sl_S1) in Sl, where RSl_S1 was 86.71 ± 11.40. On the other hand, in drought, Sl dropped to 6.86 ± 0.66 (SlD_S2) from 7.60 ± 0.71 (Sl_S2) with a relative value of 90.64 ± 6.81 in S2. Sps also witnessed a reduction in drought treatment which dropped from 43.35 ± 9.60 (Sps_S1) in S1 to 33.44 ± 7.75 (SpsD_S1) in drought. Sps_S2 was 47.15±8.18 in control, which decreased to 35.52 ± 5.73 (SpsD_S2) in drought. RSps_S1 was (78.75 ± 14.88), but was, however, higher than RSps_S2 (77.17 ± 13.78). Mean Sw_S1 was 3.05±0.65, whereas mean Sw_S2 was 2.92±0.65. Both of them decreased to 2.30 ± 0.57 (SwD_S1) and 2.25±0.41 (SwD_S2) in drought. The corresponding relative values of Sw were 77.17 ± 15.18 (RSw_S1) and 78.98 ± 14.60 (RSw_S2), respectively. Mean Tkw_S1 (48.88 ± 5.30) and mean Tkw_S2 (42.08 ± 6.26) were reduced to 45.07 ± 4.95 and 38.99 ± 5.57, respectively, in drought, whereas relative TKW values were comparable. 

### 2.5. Yield and Related Traits

Yd, biomass (Bm) and HI were included in this section. The highest decrease among all traits from control to drought was observed for Bm, which decreased from 3535.80 ± 1,086.20 (Bm_S1) in S1 and 3551.20 ± 1,172.60 (Bm_S2) in S2 to 1070.50 ± 349.64 (BmD_S1) and 1191.40 ± 373.78 (BmD_S2), respectively. In line with the same trend, Yd in S1 dropped from 1,033.10 ± 361.77 (Yd_S1) in control treatment to 350.92 ± 109.75 (YdD_S1) in drought. Likewise, Yd in S2 dropped from 1,010.10 ± 335.93 (Yd_S2) in the control treatment to 332.46 ± 116.73 in the drought treatment in S2. An increasing trend in HI was observed from control to drought in S1. For example, HI in S1 in control was 30.50 ± 10.09, which increased to 33.59 ± 7.96 (HID_S1). In S2, HI in control and drought remained almost the same. Furthermore, there was very slight increase in relative HI during both seasons (RHI_S1 = 117.66 ± 36.37, RHI_S2 = 103.30 ± 30.84). 

### 2.6. Stress Indices

Altogether, the five stress indices, that is, stress tolerance index (STI), mean productivity (MP), stress tolerance (ST), stress susceptibility index (SSI), and drought resistance index (DRI), were close to each other in both S1 and S2. However, there were differences in their ranges. For example, STI_S1 ranged between 0.06 and 1.04, whereas STI_S2 ranged between 0.06 and 1.41. Likewise, MP_S1 ranged between 266.51 and 1236.5, whereas MP_S2 ranged between 258 and 1306.5. Ranges of ST_S1 and ST_S2 were 158.19–1509 and 120–1401, respectively. Similarly, ranges of SSI_S1 and SSI_S2 were 0.44–1.32 and 0.49–1.31, respectively. Finally, DRI_S1 ranged between 49.91 and 1225.7 and DRI_S2 ranged between 49.97 and 1211.8. 

There was a significant difference (α = 0.05) observed between S1 and S2 with respect to trait values in control for Gr, Tr, Sc, Phr, Sl, Sps and Tkw (data not shown). Moreover, there was also a significant difference (α = 0.05) between S1 and S2 with respect to trait values under drought of Gr, Sc, Phr, Wue, Ht, Sps, Tkw, Bm and HI. As far as relative values are concerned, significant differences (α = 0.05) were observed between S1 and S2 for Gr, Phr, Wue, Ht, Sl and HI. However, there was no significant difference observed between the S1 and S2 trait values of all five drought indices (ST, MP, STI, SSI and DRI). 

### 2.7. Correlations

As before, the data were divided into five sections for correlation analysis (relative values were excluded). These included (1) common agronomic traits (Gr, Hd and Ph), (2) physiological traits (Tr, Sc, Phr and Wue), (3) spike-related traits (Sl, Sw, Sps, Tkw), (4) yield-related traits (Bm, Yd and HI) and (5) stress indices (STI, MP, ST, SSI and DRI). All sections were analyzed simultaneously for both S1 and S2. Data are shown in Appendix A.

### 2.8. QTL Mapping

QTL mapping analysis of several traits revealed a total of 221 (160 with LOD > 2 ≤ 3 and 61 with LOD > 3) quantitative trait loci (QTLs) were distributed on all of the fourteen durum wheat chromosomes, of which 109 (78 with LOD > 2 ≤ 3 and 31 with LOD > 3) were observed in S1 and 112 (82 with LOD > 2 ≤ 3 and 30 with LOD > 3) were observed in S2. In S1, the donor of 54 QTLs was ‘Omrabi 5’ and the donor of other 55 QTLs was ‘Belikh 2‘. Likewise, in S2, the donor of 54 QTLs was ‘Omrabi 5’ and the donor of other 58 QTLs was ‘Belikh 2’. These QTLs explained 5%–33.73% phenotypic variation in S1 and 5%–29.9% in S2. All details are given in Table 2.

From a chromosome perspective, the highest number of QTLs was observed on chromosome 5B (31 QTLs), followed by chromosomes 7B and 5A with 29 and 27 QTLs each, and by chromosomes 3B and 4B with 21 and 20 QTLs, respectively. A total of 18 QTLs was detected on chromosome 3A, whereas 15 QTLs were detected on each of the chromosomes 1B and 7A, and 14 QTLs resided on chromosome 2A. There were 10 and nine QTLs discovered on chromosomes 6B and 2B, respectively, whereas 6 QTLs detected on chromosome 6A and the least numbers of QTLs were detected on chromosomes 1A and 4A (3 QTLs each). 

#### 2.8.1. Agronomic Traits

A total of 17 QTLs were detected for Gr on 9 of the 14 chromosomes (1B (2 QTLs), 3A, 3B, 4B, 5A (2 QTLs), 5B (6 QTLs), 6A, 6B (2 QTLs) and 7B) with five of them had an LOD > 3.0. There were two QTLs common, whereas the phenotypic variation explained ranged between 6% and 18%. As far as Hd is concerned, there were 8 QTLs discovered on chromosomes 2A, 3A (2 QTLs), 5A (2 QTLs), 6B and 7B (2 QTLs), with three of them being highly significant. In addition, there was no QTL observed for Hd_S2 and RHd_S2. Altogether, QTLs explained 9%–18% of the variation in Hd. For Ph, a total of 19 (4 highly significant) QTLs were revealed on chromosomes 2A, 2B (2 QTLs), 3A (2 QTLs), 4B (6 QTLs), 5A (4 QTLs), 6A, 7A and 7B (2 QTLs) responsible for 6%–22% variation.

#### 2.8.2. Physiological Traits

There were 14 QTLs identified for Tr across both seasons on chromosomes 2A, 2B, 3A (2 QTLs), 3B, 5A, 5B (3 QTLs), 6A and 7B (4 QTLs) where two QTLs (*QTr_S2.NIAB-5B* and *QRTr_S2.NIAB-5B)* were co-located. These QTLs explained 6–14% variation. As regards to Sc, 9 QTLs were discovered on chromosomes 1B (2 QTLs), 2B, 3B, 5A, 5B, 6A (2 QTLs) and 7A responsible for 7–17% variation and one QTL (*QSc_S1.NIAB-1B.2*) was highly significant. For Phr, 17 QTLs were found on all the chromosomes except 1A and 4A. Among them, 7 QTLs were highly significant. The variation explained by these QTLs was 6–24%. There were 11 QTLs uncovered on chromosomes 2A, 2B, 4B (3 QTLs), 5A (2 QTLs), 5B, 7A and 7B (2 QTLs) for Wue from which three were highly significant, explaining 6%–22% of the phenotypic expression.

#### 2.8.3. Spike Related Traits

Among spike related traits, there were 10 QTLs residing on chromosomes 2A, 2B, 3A, 4B (2 QTLs), 5B, 7A (2 QTLs) and 7B (2 QTLs) for Sl where 4 QTLs were highly significant. These QTLs explained 7-21% variation in Sl. Likewise, 18 QTLs were found for Sw that resided on chromosomes 1A (2 QTLs), 2A (2 QTLs), 3A (3 QTLs), 3B (2 QTLs), 5A (3 QTLs), 5B (3 QTLs), 7A (2 QTLs) and 7B. With respect to Sps, 19 QTLs were discovered, which explained 6%–11% of the variation. These QTLs were lying on chromosomes 1A, 1B (3 QTLs), 3A (2 QTLs), 3B (3 QTLs), 4B, 5A, 5B (7 QTLs) and 7B. The 16 QTLs linked to Tkw were distributed on 8 different chromosomes (1B (3 QTLs), 2A, 4B (3 QTLs), 5A (3 QTLs), 5B, 6B (2 QTLs), 7A (2 QTLs) and 7B, of which only one was highly significant. All together, they explained 5%–12% of the variation. 

#### 2.8.4. Yield and Related Traits

A total of 19, 14, 16 and 12 QTLs were discovered for the yield related traits, respectively. The 19 QTLs of Bm were present on chromosomes 3A, 3B (3 QTLs), 4B, 5A, 5B (3 QTLs), 7A (3 QTLs) and 7B (7 QTLs). The 14 Yd QTLs were distributed on chromosomes 1B, 2A (3 QTLs), 3A, 3B (2 QTLs), 5A (2 QTLs), 5A, 5B, 6B (2 QTLs) and 7B (2 QTLs) explaining 6–30% of phenotypic expression. There were three chromosomes, viz., 2A, 3B and 7B, where two Yd QTLs collocated. A total of seven chromosomes (1B, 2B, 3B (3 QTLs), 4B (2 QTLs), 5B (2 QTLs), 7A and 7B (2 QTLs) carried 12 HI QTLs which explained 7–11% phenotypic differences with only one highly significant QTL. 

#### 2.8.5. Stress Indices

A total of 18 QTLs were uncovered for various stress indices viz. STI (4 QTLs), MP (6 QTLs), ST (3 QTLs) and DRI (5 QTLs) on chromosomes 1B (2 QTLs), 2A, 2B, 3B (3 QTLs), 4A (3 QTLs), 5A (5 QTLs), 5B, 6B and 7A, which provided 8–26% of the variation among various stress indices. 

## 3. Discussion

Climate change is causing more frequent and intense periods of drought as overall rainfall levels decline. Approximately 40% of the world’s land surface is characterized as dry land which inhabits 2.5 billion people, which makes up 33% of the global population [31]. Drought is currently one of the main constraints that prevent crop plants from expressing their full genetic potential. This makes the identification of drought-tolerant genotypes extremely important to secure productivity. In the present study, 114 RILs derived from a cross between Omrabi5 and Belikh2, where Omrabi5 was a drought tolerant but Bleikh2 a salt and high temperature tolerant cultivar were subjected to drought stress for two seasons to dissect drought tolerance in durum wheat. 

The genetic basis of most of the important traits in cereals is complex [32]. The same was observed in this investigation where the evaluation of 94 parameters resulted in the discovery of 221 QTLs distributed all over the durum wheat genome (Table 3). There were thirteen traits (Hd_S2, RHd_S2, TrD_S2, RSc_S1, PhrD_S1, RPhr_S1, RWue_S1, RWue_S2, RSl_S2, Sw_S2, RHI_S1, SSI_S1 and SSI_S2) for which no QTL was discovered, possibly because of the limitation of bi-parental population which is derived from two parents following fewer recombination events.

### 3.1. Phenotypic Differences between Treatments and Seasons

The different phenotypic responses to drought treatment and seasons within our RILs empowered us to genetically dissect their performance through QTL analysis. From yield perspective, yield was considerably reduced in drought in both S1 and S2. However, reduction was slightly more pronounced in S2 (RYd_S2 = 34.60 ± 12.27) as compared to S1 (RYd_S1 = 36.39 ± 12.21). This superior yield in S1 could be attributed to Sl, Sw and Sps, as these traits were higher in drought treatments in S1 than in the same treatment in S2. Seasonal difference might also be due to change in mean minimum and maximum temperatures during the two seasons as the mean minimum temperature in S1 (16.81 °C) was 1.24 °C less than mean minimum temperature in S2 (17.57 °C) although there was almost no difference on average temperatures during S1 and S2 (Appendix A). Besides, the relatively improved performance in drought in S1 of RILs could also be ascribed to 11 mm rain received in the month of Jan (when plants are heading which is sensitive to abiotic stress [33]) in S1. Superior performance in control over treatment in both S1 and S2 could also be due to prolonged growing season (~ 1 week) beside other advantages as treatment caused early flowering that reduced the assimilates time to move to plant development and growth [34].

### 3.2. Agronomic Traits

Germination plays an important role in the plant stand, which guarantees a successful harvest. In this study, the markers linked to *QGr_S1.NIAB-5A.1* (*Xbarc342*) and *QGr_S1.NIAB-6B.1* (*Xbarc178*) have been reported to be linked with initial and relative germinations, respectively [35,36], has reported one Hd QTL on the same chromosome at 58.2 ± 38.6 cM. Therefore, it can be concluded that the same QTL may be present in our study influencing Gr. Another Gr QTL (*QGr_S1.NIAB.5A.2*) is located with several other QTLs linked with Hd, Yd, ST and DRI. Six QTLs of Gr on chromosome 5B indicating that this is an important site for genes linked to plant survival. 

As regards the heading date, [29] discovered an Hd QTL at 40 cM on chromosome 2A in a durum wheat population whereas the QTL *QHdD_S1.NIAB.2A* in our present study was at 5cM on chromosome 2A. The QTL *QHd_S1.NIAB-3A* is highly comparable with the one detected by [35] for relative germination. Moreover, it is also the site of many other QTLs linked to spike related traits in this study indicating that plant development influences spike related traits and in turn final yield. It should be mentioned that a flowering time locus has been reported by [37] on chromosome 3AL. Another QTL *QHd_S1.NIAB-5A* is linked to Yd, MP and ST QTLs in this study. In addition, this QTL and *QHdD_S2.NIAB-5A* are in close vicinity to each other. Thus, the Hd QTL on chromosome 5A can be a constitutive QTL whose selection may help in yield increase as *QHdD_S2.NIAB-5A* is also linked to Yd, ST and DRI QTL in this investigation. Two Hd QTLs on chromosome 5A have been reported by [32]. Moreover, [37] discovered an Hd QTL on chromosome 5A, affirming the finding of this study. On chromosome 7B, [29] have located an Hd QTL between 0 and 44.3 cM whereas our QTL (*Q.RHd_S1.NIAB-7B*) is located at 42.5 cM confirming the findings of [29].

Most of the Ph QTLs were concentrated on chromosomes 4B and 5A. Many Ph QTLs have been reported to be located on chromosomes 1A, 1B, 1D, 2D, 3A, 4A, 4B, 6A, 7A and 7B [32,38,39]. The marker linked to *QPhD_S2.NIAB-4B and QRPh_S2.NIAB-4B* (*Xbarc193*) has been reported to be linked with coleoptile length by [40]. The same area has also been reported to carry Ph QTLs in durum wheat [39]. Likewise *QPh_S1.NIAB-6A* is highly comparable with the QTL discovered by [32] on chromosome 6A. Ref. [38] reported a QTL for Ph under drought conditions between 38 and 50 cM and another QTL under well-watered conditions at the distal end of chromosome 7AL. In addition, another QTL for Ph under drought stress has been reported to be on chromosome 7B by [38], which is linked to marker *Xbarc1073*. Our QTL *QPhD_S2.NIAB-7A* linked to marker *Xwmc517* can be highly comparable to the one reported by [38] because the marker *Xwmc517* is closely linked to *Xbarc1073* in [38]. Several Ph QTLs were co-located with QTLs of other traits. For example, *QPhD_S2.NIAB-5A* was co-located with Sc, Tr, Wue, Sw and Tkw loci. To add to this, *QRPh_S1.NIAB-7B* was co-located with a Tr QTL.

### 3.3. Physiological Traits

In our study, the QTL *QTrD_S1.NIAB-2B* for Tr may be the same as *Qgs-2B.1* (for Tr) by [38], as both lie on the proximal ends of chromosome 2BS. Another marker (*Xwmc50*) of *QRTr_S1.NIAB-3A* is reported to be linked with relative germination after experimental ageing [35] indicating the presence of stress resistance genes in this area. Ref. [38] located a QTL linked to Tr under irrigated conditions at the distal end of chromosome 5AS, whereas the QTL *QTr_S2.NIAB-5A* in our investigation was at the far end of chromosome 5AL, indicating that *QTr_S2.NIAB-5A* is a novel QTL. Similarly, the two QTLs (*QTr_S2.NIAB.5B* and *QRTr_S2.NIAB.5B*) at 158 cM linked to marker *Xgwm408* are highly comparable to one identified by [37] where the authors located a QTL linked to transpiration rate at early grain filling stage under irrigated conditions. Ref. [35] also located a Tr QTL in the distal region of chromosome 5AL. The QTL *QTrD_S1.NIAB.7B.2* is connected to marker *Xbarc276*, which is the same marker reported by [37] to be linked with Tr under irrigated conditions on chromosome 7B. 

In case of Sc, the QTL *QScD_S2.NIAB.2B* at 107.81 cM on 2B may be the same as observed by [38] at ~ 95 cM under drought stress. Ref. [37] detected several QTLs linked to leaf porosity (LP) on the distal end of chromosome 3BS. Our QTL *QScD_S2.NIAB-3B* at 13.91 cM on 3B might correspond to any of the LP QTLs of [37]. 

The 17 QTLs linked to Phr are distributed on all chromosomes except 1A, 4A and 5A, and most of them were discovered in S2 only. Nevertheless, *QPhrD_S2.NIAB-3B.1* and *QPhrD_S2.NIAB-3B.2* mirrored to a photosynthesis QTL identified by [37] under drought stress. The linked marker in both studies was *Xgwm299*. [35] located one Phr QTL under drought in the distal region of chromosome 3BS. In addition, [37] also located another Phr QTL under drought stress in the centromere region of chromosome 2B whereas in our study *QRPhr_S2.NIAB.2B* was located at the distal end of chromosome 2B. One drought and one well-watered Phr QTL by [35] and one LP QTL by [37] was located on chromosome 4B. Whether *QPhr_S1.NIAB-4B* matches any of them can only be speculated at this stage. To add to it, like in this study, [35] located Phr QTL on chromosomes 6B and 7A. However, they were at different positions. The QTL *QPhrD_S2.NIAB-7B* co-locates with normal seedling and total germination QTL of [40]. Interestingly, another QTL on 7B (*QRPhr_S2.NIAB-7B*) co-locates with Tkw, seed area and seed length by [40]. Furthermore, this QTL can also be mirrored with Phr QTLs by [41]. 

In the case of Wue, [38] located Wue QTLs at 94 cM on 2B and at ~ 238 cM on 7A, whereas *QWueD_S1.NIAB-2B* and *QWue_S1.NIAB-7A* in our study were located at the very distal ends of 2BS and 7AS, respectively, and hence are not comparable to the ones detected by [38]. However, the QTL *QWueD_S2.NIAB-7B* co-located with *QRPhr_S2.NIAB-7B* mirrored with Phr QTLs by [41]. 

### 3.4. Spike-Related Traits

With respect to Sl, *QSl_S1.NIAB-2A* mirrors with a grain protein content (GPC) QTL in [42], whereas *QSlD_S1.NIAB-2B* mirrors with Sps QTL in [43]. This QTL, however, is not the same as that described by [44], who detected consistent large effect QTLs for Sl in six environments. To add to this, *Xwmc50* and *Xgwm940a* of QTLs *QRSl_S1.NIAB-3A* and *QSlD_S2.NIAB-4B*, respectively, showed connections with germination under normal and stress conditions [35]. The QTL *QSl_S2.NIAB-5B* is also at the location of numerous QTLs linked with yield related traits in [45]. Similarly, a QTL linked to Sl has been reported by [45] on the site of *QSlD_S2.NIAB-7A.* Ref. [38] also reported a QTL linked to Sl under drought stress at 160 cM on chromosome 7A. Furthermore, two QTLs at ~ 73 cM and 148 cM have been reported to be linked with Sl in irrigated and drought conditions, respectively. We speculate the 7B QTLs of Sl (*QRSl_S1.NIAB-7B* and *QRSl_S2.NIAB-7B*) might correspond to any of those mentioned in [38]. The site of QTLs *QSwD_S1.NIAB-1A* and *QSw_S1.NIAB-1A* is reported to carry Sps QTLs [38]. Likewise, the QTLs *QSwD_S1.NIAB-2A* and *QRSw_S1.NIAB-2A* are in the region of GPC QTLs [42]. The QTLs on 3A (*QSwD_S1.NIAB-3A* and *QRSw_S1.NIAB-3A*) can be compared with relative germination QTLs [35] and Sw QTLs in [45]. The site of QTL *QSwD_S1.NIAB-3B* has been reported for Sl QTL in different environments [43]. Likewise, other QTLs have been known to be in the region of QTLs linked to yield and related traits [38]. 

Many QTLs linked to yield and spike traits have been mapped to chromosomes where Sps QTLs of this study are mapped [32,37,43]. In particular, the marker *Xwmc218* of *QSpsD_S2.NIAB-7B* has been reported to be in connection with seed area and length in [40], providing a clear connection towards yield. Moreover, *QRSps_S1.NIAB* in this study is located at 104 cM mirrored with grains per spike QTL of [44] who reported the Sps QTL at 102.5 cM.

Four QTLs among 16 of Tkw were co-located with other Tkw QTLs. The 1B loci (*QRTkw_S.NIAB-1B*, *QTkwD_S1.NIAB-1B.1* and *QTkw_S1.NIAB.2*) can be compared with the Yd QTLs of [38] and meta-QTL of Yd by [43]. Furthermore, *QRTkw_S.NIAB-1B, QTkw_S2.NIAB-4B, QRTkw_S2.NIAB-4B, QTkDw_S1.NIAB-5A, QRTkw_S1.NIAB.5A and QRTkw_S1.NIAB.7B* mirror relative germination QTLs by [35]. 

### 3.5. Yield and Related Traits

Chromosome 1B is an important site of yield QTLs as several yield QTLs on this chromosome in hexaploid [29] or durum wheat [38,46]. Thus, the QTL *QRYd_S2.NIAB-1B* could be one reported previously. Interestingly, this QTL also co-locates with coleoptile length under drought [40] in the same population, pointing towards stress tolerance genes in this area. Similarly, there were three QTLs for yield (*QYdD_S1.NIAB-2A*, *QYdD_S2.NIAB-2A* and *QRYd_S1.NIAB-2A*) on chromosome 2A where *QYdD_S2.NIAB-2A* and *QRYd_S1.NIAB-2A* were linked to the same marker (*Xgwm895*). Consistent QTLs for Yd have been detected on chromosome 2A [43,45]. The QTL *QYdD_S1.NIAB-2A* can be compared with 2AS Yd QTLs of [42], whereas the other two can be compared with the QTLs on 2AL. Ref. [29] also detected a Yd QTL on 2A linked to marker *Xwmc177*, which is closely linked to *Xgwm726* (related to *QYdD_S1.NIAB-2A*). The 3A QTL (*QYd_S2.NIAB-3A*) seems not to be comparable to those observed by [29] in durum wheat and by [36] in hexaploid wheat, because their QTLs were either in the centromere region or at the distal end of 3AL whereas our QTL was at 52.11 cM. However, this QTL could be compared Yd QTL described by [34], who described a cluster of QTLs including Yd QTL between 41.3 and 51.4 cM in durum wheat. Moreover, [46] have also located a Yd QTL under drought stress at 62.9 cM. To add to it, *QYd_S2.NIAB-3A* does co-locate with a grain number QTL of [32]. The QTLs viz. *QYd_S2.NIAB-3B* and *QRYd.NIAB-3B* linked to marker *Xgwm547* are at the tip of long arm of 3B. Consistent QTLs for Yd under heat, drought and irrigated conditions have been described by [34] at 212 cM on chromosome 3BL indicating that our QTL could be the same as that of [34]. Likewise, both [29] and [45] have located QTLs for Yd at the same location, whereas [36] located a Tkw QTL in hexaploid wheat in this area indicating the connection of Yd with Tkw. One of the two 5A QTLs (*QYd_S1.NIAB-5A* and *QRYd_S2.NIAB-5A*) co-locates with a germination QTL in [35], whereas [32] located several Sl, Sw and Tkw QTL on the entire length of chromosome 5A, suggesting some connection between yield and these components. In addition, [32] have reported numerous QTLs linked to yield and related traits on chromosome 5A [47]. Furthermore, both [32] and [47] have reported Yd QTLs on the proximal end of chromosome 5BS, whereas *QYdD_S1.NIAB.5B* lies at 143.21 cM, and so cannot be compared with either of the above-mentioned investigations. Ref. [29] located a yield locus exactly in the same area as that of *QRYd_S1.NIAB.6B*. Nevertheless, [32] mapped numerous Yd related QTL on chromosome 6B. Chromosome 7B has been reported to be an important site for Yd and Yd related traits by [32], and [29] located a stable QTL for Yd between 19 and 33 cM and one QTL restricted to one environment between 106 and 122 cM. This study linked *QYd_S1.NIAB.7B* and *QYdD_S2.NIAB.7B* (216-226 cM) with *Xbarc276*, which is in between *Xgwm333* and *Xbarc176*. This interval is linked with a Yd QTL in [29]. 

There are no investigations exclusively reporting Bm QTLs except [38] and [48]. No QTLs of Bm in the present study match with those reported by [38] and [48]. However, the QTL *QBmD_S1.NIAB-7A* of 7A is very near to GPC and Sl QTL as reported by [32] in hexaploid wheat. Likewise, the *QRBm_S2.NIAB-7B* of 7B is linked with QTLs of seed area and seed length [41]. Among other traits, the Bm QTLs co-locate with Gr, Ph, HI, Sw, Sps, Yd and HI QTLs in this population indicating that these are the sites of important genes.

The importance of HI in relation to drought has been indicated recently [36,44]. None of the 12 HI QTLs in this study matched with those of [38], who located HI QTLs in hexaploid wheat on chromosomes 1A, 2A, 2B, 3A, 4A, 4D and 6A. Numerous QTLs for HI have been identified on chromosomes 1B, 5B, 5D 6A, 7A, 7B and 7D using an association mapping approach and a different marker system [49]. The 1B QTL of HI (*QRHI_S1.NIAB-1B*) in this study was located at 93.11 cM whereas the QTL in [49] was located at 158 cM; hence, they seem not to be comparable with each other. *QRHI_S1.NIAB-1B* also co-locates with coleoptile length under drought stress in [41]. The QTL *QHID_S.NIAB-5B* on 5B can be the same as that of [49] who discovered an HI QTL under non stress conditions. The chromosome 7B carried QTLs for HI in non-stress, drought, heat and combined heat and drought stress most notable at 163 cM in [49], whereas our QTLs *QRHI_S2.NIAB-7B* and *QHI_S2.NIAB-7B* were at a distance of 184.41 and 336.01 cM, respectively.

### 3.6. Stress Indices

Several tolerance indices have been devised and used by breeders to identify the best drought-tolerant lines. It has been suggested that each index may be a potential indicator of different biological responses to drought [31]. STI, MP and geometric mean productivity have been concluded to be effective indices in discriminating drought tolerance genotypes in durum wheat [50]. Chromosome 2B has been an important site harboring STI QTLs under drought, heat and combined drought and heat stress [49]. However, our QTL *QDRI_S2.NIAB-2B* does not correspond to any of those QTLs because of a different position. Chromosome 4A also carried STI QTLs under combined stress at 67.71 and 71.47 cM in [49] indicating some similarity with our 4A QTLs which were at 44.51 (*QSTI_S2.NIAB-4A.1*), 64.31 (*QDRI_S1.NIAB-4A*) and 100.81 (*QSTI_S2.NIAB-4A.2*) cM. To add to it, [50] also located various QTLs linked with yield and yield related stress indices on chromosome 4A. The marker *Xgwm299* on 3B in association with *QMP_S2.NIAB-3B.1* and *QST_S2.NIAB-3B* co-locates with Phr, Sw and Sps QTLs. Furthermore, the same marker is linked with Tkw [51] and Phr at late growing stages [40] in other studies. Furthermore, the interval of *QDRI_S1.NIAB-5B* (*Xwmc28-Xgwm1072*) is linked with HI, spike HI and spike number QTLs in durum wheat [52]. 

### 3.7. Co-Location QTLs and MAS for Durum Wheat Breeding

From the total of 221 QTLs, 145 could be clustered (2 or more QTLs linked to the same marker) into a total of 53 clusters (Figure 1), covering all the durum wheat chromosomes except 4A. The number of QTLs clustered together ranged between two and six. Among them, there were at least 20 markers that were associated with any trait in both seasons, whereas 10 markers were associated with the same trait under various conditions in a single season. Moreover, there were four markers that were associated with the same trait under different conditions in both seasons. It is also evident that certain Yd QTLs match exactly with Gr (cluster 5A.5), Tr (cluster 7B.5), Hd (cluster 5A.4, 5A.5), Tkw (clusters 1B.1, 2A.1), HI (cluster 3B.5), Sps (cluster 5B.2) and stress indices (clusters 5A.4, 5A.5) QTLs. Clusters of QTLs have also been reported previously [32,34,38,45,47], indicating that multiple genes might be present at such sites which can either be constitutive or expressive. Plant breeders, however, tend to look for stable loci to improve yield. Although several QTLs can be targeted for yield enhancement, the most promising one could be the clusters 2A.4 and 7B.5, where markers *Xgwm895* and *Xbarc276* were associated with Yd in both S1 and S2. The allelic profile of yield QTLs in these clusters indicate that allele A of *Xgwm895* and allele B of *Xbarc276* can enhance the Yd up to 6.16% in control and 5.27% in drought (Table 4). Moreover, if combined, a yield gain of up to 11% is possible. 

## 4. Materials and Methods

A set of 114 RILs was developed from a cross between a drought-tolerant cultivar Omrabi5 (P1) and a high temperature- and salinity-tolerant breeding line Belikh2 (P2) [17] at ICARDA, Syria, by repeated selfing of F_1_, using the single seed descent method [28]. Seeds from the 2015 season in Germany were multiplied in 2015-16 at NIAB, Pakistan, and were subsequently used for the investigation in 2016-17 and the seeds of the 2016-17 harvest were used in 2017-18. 

### 4.1. Experimental Design and Traits Measured

A total of two experiments were conducted to investigate the population with respect to yield and other related traits. They were conducted in 2016-17 (S1) and 2017-18 (S2) seasons at NIAB, Pakistan, at exactly the same experimental site to avoid any soil interaction. Moreover, all the data were collected at almost the same dates in both experiments to minimize the effect of environment on phenotypic expression of the measured traits. The population was sown on 1st of November each season with one replicate in control and three under drought in both S1 and S2 cycles in the form of two rows per RIL. Seeds were sown manually with 5g seeds/row, in 2-meter rows using hand-driven drill and row to row distance was kept 20 cm. Control plots were irrigated four times (15th Nov., 1st Jan., 15th Feb. and 15th March), whereas drought plots were kept devoid of any irrigation. The drought and control experiments were in the same field, where the control experiment was separated from the drought by making a 2-ft-high ridge partition and there was a gap of at least 5 meters between drought and control treatments. Furthermore, there was also a gap of 5 meters between drought treatment and water channel to avoid any leach down effect that might have an effect on drought treatments. To add to it, we planted brassica in the leftover area so that the leached water from the water channel and control experiment is used up before it reached to the drought treatment. The total rainfall in S1 was 36.25 days, distributed over the period of 14 days (from Nov 2016-May 2017), where the maximum rain was received in April 2017 (16.58 mm). In S2, total rainfall was 42.48, distributed over a period of 30 days with maximum rainfall in April 2018 (Appendix A). Furthermore, temperature data is provided in Appendix A and layout plan of the experiments is provided in Appendix A. A list of parameters recorded during the course of the two experiments is given in Table 4. Correspondingly, relative values were also calculated by dividing drought by the control times one hundred.

#### 4.1.1. Agronomic Traits

Germination % (Gr/GrD) was recorded visually three weeks after sowing according to a scale ranging from 0%–100%, where 0% = no germination and 100% = complete germination. Heading time (Hd/HdD) was measured when 50% of spikes have left the flag leaf. Plant height (Ph/PhD) was recorded in cm from five random plants per replicate, with the help of meter rod, measured from the base of the plant to the spike tip excluding awns, before harvest.

#### 4.1.2. Physiological Traits

Transpiration rate (Tr/TrD) was calculated using the handheld porometer (Model: LI-1600 steady state porometer) in µg cm^−2^s^−1^. Data were recorded separately on cloudless days for control (on 99th day of sowing) and drought (on 100th and 101st day of sowing) between 11:00 am and 2:00 pm. Each reading was a mean of three randomly selected flag leaves per replicate. Stomatal conductance (Sc/ScD) (mmol m^−2^s^−1^) was calculated using the formula: 1/DR × CF where DR is diffusible resistance calculated from porometer, CF = correction factor which is calculated using the formula: LT × constant where the value of constant at 25 °C for durum was 411.8, and LT was leaf temperature calculated from porometer. Photosynthetic rate (Phr/PhrD) was determined from Sc and Tr using the formula: Phr = Sc (mmol m^−2^s^−^/Tr (µg cm^−2^s^−1^) × 10. Water use efficiency (Wue/WueD) was calculated by dividing the corresponding photosynthetic rate with transpiration rate times 100.

#### 4.1.3. Spike-Related Traits

For spike-related traits, five spikes per row per replicate were collected randomly from the two experiments whereas for other traits, values from random plants were taken. Spike length (Sl/SlD) was calculated by taking mean length of five spikes/replicate using scale in centimeters. Spike weight (Sw/SwD) in grams was calculated by taking mean weight of five spikes using electronic weighing balance. The number of seed per spike (Sps/SpsD) was calculated by threshing and counting the seeds from five random spikes per replicate. Thousand kernel weight (Tkw/TkwD) in grams was recorded by weighing 1000 grains using electronic balance. 

#### 4.1.4. Yield and Related Traits

These included biomass (Bm), yield (Yd) and harvest index (HI). Biomass (Bm/BmD) (g) was measured by weighing the complete harvest of above ground part excluding roots per plot where harvesting was done manually to avoid any mixing of the straw. Yield (Yd/YdD) was recorded after manual threshing in grams per plot. Harvest index (HI/HID) was determined by dividing the corresponding Yd with respective Bm.

#### 4.1.5. Stress Indices

Five stress indices were calculated using the yield data. These included stress tolerance index (STI), mean productivity (MP), stress tolerance (ST), stress susceptibility index (SSI), and drought resistance Index (DI), whose formulas are provided in Table 4.

### 4.2. Data Analysis

Statistica software was used for descriptive statistics of the measured traits and to construct box plots. ANOVA was performed using the RStudio-1.0.153.exe program.

### 4.3. Genotyping and Genetic Mapping

The genetic map of Omrabi5 × Belikh2 RIL population comprising 265 microsatellite loci, spanning 2864 cM (10.8 cM mean inter-marker separation) and covering all the 14 chromosomes of durum wheat, was available from [40]. The individual chromosomes ranged in genetic length from 37 cM (chromosome 1A) to 366 cM (chromosome 7B). The 265 microsatellites comprised 159 GWM, 62 BARC and 44 WMC markers. Further details about the map are available in [40]. QTLs were assigned using a composite interval mapping method (model 6 with forward stepwise regression) implemented in QTL *Cartographer v2.5* [53]. To control the effect of genetic background, five markers, identified by forward regression, were used as a cofactor with a window size of 5cM. A LOD score of >2.0≤ 3 was applied to detect QTLs as significant and >3.0 as highly significant. 

## 5. Conclusions

In the present study, we genetically dissected the performance of durum wheat RIL population for yield and yield-related traits and various morpho-physiological characteristics under irrigated and drought conditions through QTL mapping, which was mapped with the 265 assays. The experiments were conducted in two consecutive seasons at the exact same location to avoid variation due to soil properties and ground factor. A total of 221 QTLs were identified, where 145 QTLs were co-located with each other, and where 20 markers were associated with any trait in either S1 or S2. Furthermore, certain Yd QTLs match exactly with Gr, Tr, Hd, Tkw, HI, Sps and stress index QTLs, a finding consistent with previous reports. Several QTLs can be targeted for yield enhancement in durum wheat; the most promising ones from this study could be the clusters 2A.4 and 7B.5, where markers *Xgwm895* and *Xbarc276*, if combined, might increase the yield up to 11%. In addition, an indirect improvement in yield is also possible by improving other traits (Tr, Tkw and HI), as we have found co-located QTLs for the said traits with Yd. Meanwhile, more studies are needed to validate our findings, our study still provides new insights into the genetic architecture of yield and related traits in durum wheat and the RILs with favorable loci could be used to improve yield in durum wheat.

## Figures and Tables

**Figure 1 ijms-21-02372-f001:**
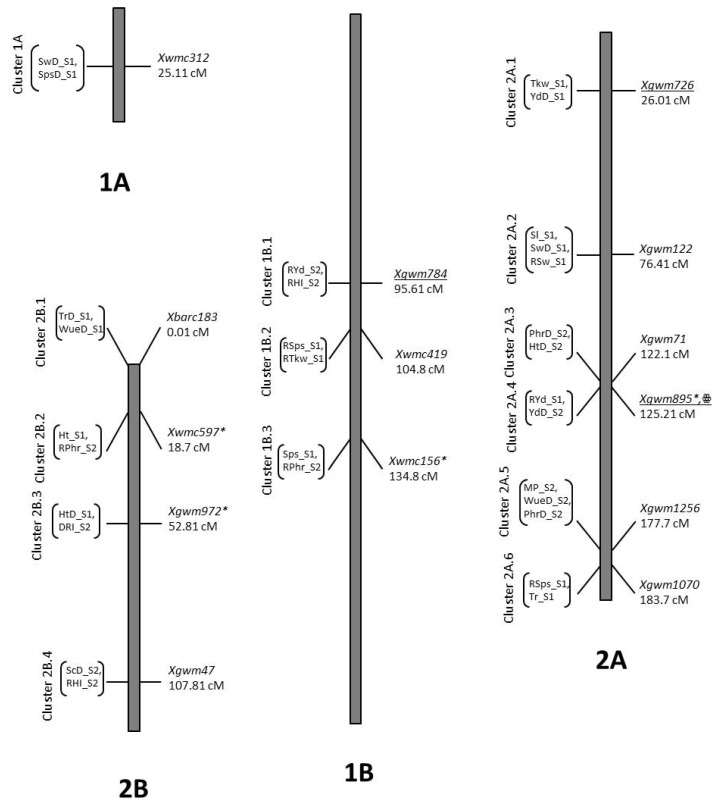
Distribution of QTL clusters in RIL population of durum wheat population. * = marker linked with a QTL in both seasons for any trait, ** = marker linked with the same trait in different conditions in a single season, ꙮ = marker linked with the same trait either in different conditions in both seasons. Markers linked with any yield QTL are under lined. For abbreviation, refer to Table 4.

**Table 1 ijms-21-02372-t001:** Range and mean ± SD of traits measured in S1 and S2 including significant differences among genotypes, seasons, treatments, genotype x season, genotype x treatment, season x treatment and genotype x season x treatment interactions at *p*-values 0.05 (*), 0.01 (**) and 0.001 (***) in durum wheat RIL population. Gr = germination %, Tr = Transpiration rate, Sc = Stomatal conductance, Phr = Photosynthetic rate, Wue = water use efficiency, Hd = heading time, Ph = plant height, Sl = spike length, Sw = spike weight, Sps = seeds per spike, Bm = biomass, Yd = yield, HI = harvest index, Tkw = thousand kernel weight, STI = stress tolerance index, MP = mean productivity, ST = stress tolerance, SSI= stress susceptibility index, DRI = drought resistance index, S1 = season 1 (2016-17), S2 = (2017-18), D = drought, R = relative and NA = not applicable.

S1			S2			*p*-Value
Trait	Range	Mean ± SD	Trait	Range	Mean ± SD	
**Agronomic Traits**						
Gr_S1	50–100	81.86 ± 9.63	Gr_S2	80–100	96.00 ± 3.72	**P_S_ = *****
GrD_S1	47.5–100	72.55 ± 9.96	GrD_S2	70–100	91.24 ± 5.56	**P_T_ = *****
RGr_S1	66.7–100	88.24 ± 11.78	RGr_S2	73.68–100	95.11 ± 5.30	**P_G_ = ***
						**P_S*T_ = ***
Hd_S1	80–128	100.47 ± 11.79	Hd_S2	84–120	98.31 ± 7.19	**P_T_ = *****
HdD_S1	80–126	93.94 ± 12.41	HdD_S2	78.5–114.5	92.32 ± 6.79	**P_S*T_ = ***
RHd_S1	51–121.78	94.30 ± 3.38	RHd_S2	80–101.66	94.18 ± 2.70	
Ht_S1	61–148	114.78 ± 23.82	Ht_S2	62–185	111.36 ± 26.31	**P_S_ = *****
HtD_S1	53–110	80.24 ± 12.45	HtD_S2	51–109.83	83.65 ± 14.14	**P_T_ = *****
RHt_S1	45.07–99.1	74.14 ± 10.05	RHt_S2	56.15–98.03	76.51 ± 8.94	
**Physiological Traits**						
Tr_S1	22.55–56.02	37.38 ± 6.86	Tr_S2	20.02–48.25	34.17 ± 6.51	**P_T_ = *****
TrD_S1	5.17–27.8	15.56 ± 4.65	TrD_S2	8.59–19.68	15.06 ± 1.90	**P_G_ = ****
RTr_S1	14.2–95.5	42.98 ± 15.98	RTr_S2	21.28-73.24	45.43 ± 9.78	**P_T*G_ = ***
Sc_S1	675.08–3743.6	1492.70 ± 527.26	Sc_S2	526.72–2059.5	1143.90 ± 293.57	**P_S_ = *****
ScD_S1	118.33–675.08	342.96 ± 114.03	ScD_S2	99.85–486.87	262.53 ± 93.35	**P_T_ = *****
RSc_S1	3.16–61.2	25.29 ± 11.39	RSc_S2	10.14–47.19	23.50 ± 8.17	**P_G_ = *****
						**P_S*T_ = ****
						**P_T*G_ = ****
						**P_S*T_ = ***
Phr_S1	2.5–10.2	3.97 ± 1.24	Phr_S2	1.79–8.6	3.49 ± 1.25	**P_S_ = *****
PhrD_S1	1.25–3.84	2.22 ± 0.44	PhrD_S2	0.57–3.78	1.72 ± 0.67	**P_T_ = *****
RPhr_S1	14.93–93	58.96 ± 15.51	RPhr_S2	15.95–97.81	50.94 ± 15.76	**P_G_ = ***
						**P_S*T_ = ****
Wue_S1	7.32–30.59	11.89 ± 4.03	Wue_S2	4.41–36.12	11.14 ± 6.30	**P_S_ = ****
WueD_S1	5.95–54.63	16.04 ± 7.93	WueD_S2	3.29–31.33	11.43 ± 5.14	**P_T_ = *****
RWue_S1	30.56–659.29	144.41 ± 87.62	RWue_S2	24.38–225.63	113.98 ± 41.44	**P_S*T_ = ****
**Spike Related Traits**						
Sl_S1	6.4–12.06	8.25 ± 0.96	Sl_S2	5.8–9.96	7.60 ± 0.71	**P_S_ = ***
SlD_S1	4.43–9.58	7.13 ± 1.11	SlD_S2	4.4–9	6.86 ± 0.66	**P_T_ = *****
RSl_S1	56.51–99.43	86.71 ± 11.40	RSl_S2	74.48–100	90.64 ± 6.81	**P_S*T_ = *****
						**P_T*G_ = ***
Sps_S1	24.6–62.6	43.35 ± 9.60	Sps_S2	32.8–91.3	47.15 ± 8.18	**P_T_ = *****
SpsD_S1	16.6–56.6	33.44 ± 7.75	SpsD_S2	15.8–48.1	35.52 ± 5.73	
RSps_S1	43.24–99.44	78.75 ± 14.88	RSps_S2	34.35–99.75	77.17 ± 13.78	
Sw_S1	1.45–4.94	3.05 ± 0.65	Sw_S2	1.58–4.96	2.92 ± 0.65	**P_T_ = *****
SwD_S1	1.19–3.76	2.30 ± 0.57	SwD_S2	1.28–3.66	2.25 ± 0.41	
RSw_S1	42.41–100	77.19 ± 15.48	RSw_S2	43.2–99.66	78.98 ± 14.60	
Tkw_S1	27.52–61.94	48.88 ± 5.30	Tkw_S2	31.09–56.4	42.08 ± 6.26	**P_T_ = *****
TkwD_S1	26.4–55.21	45.07 ± 4.95	TkwD_S2	26.53–54.87	38.99 ± 5.57	**P_G_ = ****
RTkw_S1	67.96–99.91	92.37 ± 6.66	RTkw_S2	69.54–99.93	92.20 ± 5.66	
**Yield and Related Traits**						
Bm_S1	1180.8–6266.5	3535.80 ± 1086.20	Bm_S2	1120–9000	3551.20 ± 1172.60	**P_T_ = *****
BmD_S1	453.6–2282.6	1070.50 ± 349.64	BmD_S2	420–2320	1191.40 ± 373.78	**P_S*G_= ***
RBm_S1	10.03–92.01	32.15 ± 12.00	RBm_S2	12–62.42	34.60 ± 11.05	
Yd_S1	305.68–1882.7	1033.10 ± 361.77	Yd_S2	306–1821	1010.10 ± 335.93	**P_T_ = *****
YdD_S1	120.73–639.19	350.92 ± 109.75	YdD_S2	138–792	332.46 ± 116.73	**P_G_ = ***
RYd_S1	12.5–70.93	36.39 ± 12.21	RYd_S2	11.92–67.08	34.89 ± 12.27	**P_S*G_= *****
						**P_S*T*G_=****
HI_S1	11.58–66.6	30.50 ± 10.09	HI_S2	15.65–46.27	28.56 ± 6.38	**P_T_ = *****
HID_S1	14.16–60.49	33.59 ± 7.96	HID_S2	12.38–44	28.47 ± 6.68	**P_G_ = *****
RHI_S1	40.55–219.08	117.66 ± 36.37	RHI_S2	41.31–250.31	103.30 ± 30.84	**P_T*G_ =*****
**Stress Indices**						
STI_S1	0.06–1.04	0.36 ± 0.20	STI_S2	0.06–1.41	0.35 ± 0.21	NA
MP_S1	266.51–1236.5	693.77 ± 213.97	MP_S2	258–1306.5	674.98 ± 198.37	NA
ST_S1	158.19–1509	685.70 ± 322.29	ST_S2	120–1401	684.24 ± 297.06	NA
SSI_S1	0.44–1.32	0.96 ± 0.18	SSI_S2	0.49–1.31	0.97 ± 0.18	NA
DRI_S1	49.91–1255.7	389.18 ± 208.56	DRI_S2	49.97–1211.8	371.11 ± 229.50	NA

**Table 2 ijms-21-02372-t002:** Composite interval mapping analysis of the durum wheat RIL mapping population. Highly significant LODs are in bold. QTL with similar superscripts are likely identical loci.

Trait	QTL Designation	Chr	Pos	LOD	Closest Marker	Add. Effect	R^2^	Donor
**Agronomic Traits**								
Gr_S2	*QGr_S2.NIAB-1B.1*	1B	19.81	**3.75**	*Xgwm752*	−1.75	0.18	B2
Gr_S2	*QGr_S2.NIAB-1B.2*	1B	31.01	**5.41**	*Xbarc8*	−1.77	0.18	B2
Gr_S2	*QGr_S2.NIAB-3A^s^*	3A	96.91	2.23	*Xbarc356*	1.05	0.07	O5
GrD_S2	*QGrD_S2.NIAB-3B^t^*	3B	11.91	2.51	*Xgwm285*	1.59	0.07	O5
GrD_S1	*QGrD_S1.NIAB-4B^aa^*	4B	36.51	2.04	*Xgwm710b*	−2.64	0.06	B2
Gr_S1	*QGr_S1.NIAB-5A.1*	5A	73.71	2.51	*Xbarc342*	3.06	0.08	O5
Gr_S1	*QGr_S1.NIAB-5A.2^hh^*	5A	171.81	**3.59**	*Xgwm1171b*	−4.27	0.18	B2
GrD_S1	*QGrD_S1.NIAB-5B*	5B	48.71	2.81	*Xgwm1165*	−3.31	0.1	B2
GrD_S2	*QGrD_S2.NIAB-5B.1^nn^*	5B	207.51	2.6	*Xbarc266*	−1.71	0.08	B2
RGr_S2	*QRGr_S2.NIAB-5B.1^nn^*	5B	207.51	2.87	*Xbarc266*	−2.02	0.1	B2
GrD_S2	*QGrD_S2.NIAB-5B.2*	5B	247.11	2.95	*Xbarc59*	−1.69	0.09	B2
GrD_S2	*QGrD_S2.NIAB-5B.3^pp^*	5B	256.01	2.03	*Xgwm790*	−1.4	0.06	B2
RGr_S2	*QRGr_S2.NIAB-5B.2^pp^*	5B	261.01	2.31	*Xgwm790*	−1.76	0.09	B2
Gr_S2	*QGr_S2.NIAB-6B*	6A	121.51	**3.09**	*Xgwm4915*	1.44	0.14	O5
Gr_S1	*QGr_S1.NIAB-6B.1^rr^*	6B	141.41	2.9	*Xbarc178*	−3.64	0.12	B2
Gr_S1	*QGr_S1.NIAB-6B.2*	6B	147.11	**3.02**	*Xbarc247*	−3.53	0.11	B2
RGr_S1	*QRGr_S1.NIAB-7B*	7B	216.21	2.25	*Xgwm983*	−3.69	0.08	B2

HdD_S1	*QHdD_S1.NIAB-2A*	2A	5.81	2.21	*Xgwm1244*	−4.23	0.08	B2
Hd_S1	*QHd_S1.NIAB-3A^q^*	3A	73.51	**4.54**	*Xwmc50*	−5.11	0.17	B2
HdD_S1	*QHdD_S1.NIAB-3A*	3A	44.51	2.1	*Xgwm779*	−4.47	0.09	B2
Hd_S1	*QHd_S1.NIAB-5A^gg^*	5A	169.81	2.52	*Xbarc165*	−4.36	0.12	B2
HdD_S2	*QHdD_S2.NIAB-5A^hh^*	5A	188.01	**3.39**	*Xgwm1171b*	−3.02	0.18	B2
RHd_S1	*QRHd_S1.NIAB-6B*	6B	123.6	2.35	*Xgwm963a*	2.5	0.1	O5
RHd_S1	*QRHd_S1.NIAB-7B*	7B	42.5	**4.35**	*Xgwm195*	4.02	0.16	O5
RHd_S1	*QRHd_S1.NIAB-7B^ccc^*	7B	223.8	2.05	*Xwmc396*	−2.3	0.08	B2

PhD_S2	*QPhD_S2.NIAB-2A^h^*	2A	121.31	2.04	*Xgwm71*	3.64	0.06	O5
Ph_S1	*QPh_S1.NIAB-2B^l^*	2B	19.71	2.18	*Xwmc597*	6.88	0.07	O5
PhD_S1	*QPhD_S1.NIAB-3A*	3A	14.61	2.86	*Xbarc294*	−3.99	0.09	B2
PhD_S1	*QPhD_S1.NIAB-2B^m^*	2B	52.81	2.29	*Xgwm972*	−3.41	0.06	B2
Ph_S2	*QRPh_S2.NIAB.3A*	3A	13.01	2.21	*Xbarc57*	2.39	0.06	B2
Ph_S2	*QRPh_S2.NIAB.4B^y^*	4B	9.91	2.8	*Xbarc193*	−2.66	0.08	B2
PhD_S2	*QPhD_S2.NIAB-4B.1^y^*	4B	14.91	2.92	*Xbarc193*	4.78	0.1	O5
PhD_S1	*QPhD_S1.NIAB-4B^z^*	4B	15.91	**5.2**	*Xgwm925*	6.15	0.22	O5
Ph_S2	*QPh_S2.NIAB-4B.1^z^*	4B	24.01	2.21	*Xgwm925*	8.37	0.09	O5
PhD_S2	*QPhD_S2.NIAB-4B.2^z^*	4B	25.01	2.51	*Xgwm925*	4.69	0.1	O5
Ph_S2	*QPh_S2.NIAB-4B.2^aa^*	4B	35.71	2.99	*Xgwm710b*	8.58	0.1	O5
Ph_S1	*QPh_S1.NIAB-5A^ee^*	5A	46.91	2	*Xgwm129*	6.04	0.06	O5
PhD_S2	*QPhD_S2.NIAB-5A^ii^*	5A	277.71	2.3	*Xgwm126*	−4.36	0.09	B2
Ph_S2	*QPh_S2.NIAB-5A^jj^*	5A	334.41	2.42	*Xbarc261*	−7.91	0.08	B2
RPh_S1	*QRPh_S1.NIAB-5A*	5A	311.71	**3.3**	*Xgwm995*	3.68	0.1	O5
Ph_S1	*QPh_S1.NIAB-6A*	6A	51.31	**3.16**	*Xgwm4608*	−8.31	0.1	B2
PhD_S2	*QPhD_S2.NIAB-7A*	7A	144.21	2.04	*Xwmc488*	3.68	0.06	O5
RPh_S1	*QRPh_S1.NIAB-7B^bbb^*	7B	252.61	2.92	*Xgwm112*	3.84	0.09	O5
RPh_S2	*QRPh_S2.NIAB.7B*	7B	288.81	**4.78**	*Xwmc517*	4.73	0.16	O5
**Physiological Traits**								
Tr_S1	*QTr_S1.NIAB-2A*	2A	189.71	2.8	*Xgwm382*	−2.41	0.11	B2
TrD_S1	*QTrD_S1.NIAB-2B^k^*	2B	0.01	**4.13**	*Xbarc183*	1.9	0.12	O5
RTr_S1	*QRTr_S1.NIAB-3A^q^*	3A	73.51	**3.49**	*Xwmc50*	5.87	0.13	O5
TrD_S1	*QTrD_S1.NIAB-3A^r^*	3A	77.51	**3.48**	*Xgwm5*	1.74	0.13	O5
RTr_S2	*QRTr_S2.NIAB.3B*	3B	152.31	2.2	*Xgwm655*	−11.61	0.14	B2
Tr_S2	*QTr_S2.NIAB.5A^ii^*	5A	274.71	2.41	*Xgwm126*	−1.1	0.07	B2
TrD_S1	*QTrD_S1.NIAB-5B^kk^*	5B	13.41	2.24	*Xgwm540a*	−1.24	0.06	B2
RTr_S2	*QRTr_S2.NIAB.5B^mm^*	5B	158.91	2.03	*Xgwm408*	−8.41	0.06	B2
Tr_S2	*QTr_S2.NIAB.5B^mm^*	5B	158.11	2.9	*Xgwm408*	1.34	0.09	O5
Tr_S2	*QTr_S2.NIAB.6A^qq^*	6A	246.21	2.55	*Xwmc621*	1.45	0.12	O5
TrD_S1	*QTrD_S1.NIAB-7B*	7B	51.11	**3.07**	*Xgwm400*	1.68	0.11	O5
TrD_S1	*QTrD_S1.NIAB-7B.2^aaa^*	7B	219.11	2.48	*Xbarc276*	−1.31	0.07	B2
RTr_S1	*QRTr_S1.NIAB-7B^ccc^*	7B	238.91	2.19	*Xwmc396*	5.65	0.11	O5
Tr_S1	*QTr_S1.NIAB-7B^bbb^*	7B	248.91	**3.03**	*Xgwm112*	−2.69	0.14	B2

Sc_S1	*QSc_S1.NIAB-1B.1*	1B	168.61	2.47	*Xbarc61*	168.28	0.08	O5
Sc_S1	*QSc_S1.NIAB-1B.2*	1B	192.41	**5.03**	*Xwmc134*	−245.34	0.17	B2
ScD_S2	*QScD_S2.NIAB-2B^n^*	2B	107.81	2.17	*Xgwm47*	1.81	0.07	O5
ScD_S2	*QScD_S2.NIAB-3B^u^*	3B	13.91	2.96	*Xgwm685*	−2.18	0.09	B2
RSc_S2	*QRSc_S2.NIAB-5A^ii^*	5A	272.61	2.41	*Xgwm126*	10.03	0.1	O5
Sc_S2	*QSc_S2.NIAB-5B^mm^*	5B	158.11	2.83	*Xgwm408*	4.29	0.09	O5
Sc_S2	*QSc_S2.NIAB-6A.1*	6A	80.01	2.01	*Xbarc107*	−3.28	0.05	B2
Sc_S2	*QSc_S2.NIAB-6A.2^qq^*	6A	244.21	2.44	*Xwmc621*	4.88	0.13	O5
ScD_S1	*QScD_S1.NIAB-7A^tt^*	7A	0.01	2.92	*Xgwm1171a*	37.93	0.09	O5

RPhr_S2	*QRPhr_S2.NIAB-1B^d^*	1B	140.81	**3.65**	*Xwmc156*	6.26	0.15	O5
PhrD_S2	*QPhrD_S2.NIAB-2A.1^h^*	2A	121.31	**3.15**	*Xgwm71*	0.14	0.1	O5
PhrD_S2	*QPhrD_S2.NIAB-2A.2^j^*	2A	175.21	**5.41**	*Xgwm1256*	−0.25	0.24	B2
RPhr_S2	*QRPhr_S2.NIAB-2B^l^*	2B	18.71	2.25	*Xwmc597*	4.63	0.07	O5
RPhr_S2	*QRPhr_S2.NIAB-3A.1^o^*	3A	17.61	**3.68**	*Xgwm757*	−5.44	0.12	B2
Phr_S2	*QPhr_S2.NIAB-3A^r^*	3A	77.51	**4.5**	*Xgwm5*	0.12	0.17	O5
RPhr_S2	*QRPhr_S2.NIAB-3A.2*	3A	108.01	2.36	*Xgwm666*	−4.58	0.07	B2
PhrD_S2	*QPhrD_S2.NIAB-3B.1^w^*	3B	181.81	**3.75**	*Xgwm299*	−0.15	0.13	B2
PhrD_S2	*QPhrD_S2.NIAB-3B.2^w^*	3B	186.81	2.82	*Xgwm299*	−0.14	0.1	B2
Phr_S1	*QPhr_S1.NIAB-4B^bb^*	4B	43.31	2.34	*Xgwm1167*	−0.34	0.07	B2
Phr_S2	*QPhr_S2.NIAB-5B*	5B	6.41	**4.5**	*Xgwm191*	0.09	0.14	O5
Phr_S1	*QPhr_S1.NIAB-6A*	6A	2.01	2.15	*Xgwm459*	−0.31	0.06	B2
Phr_S1	*QPhr_S1.NIAB-6B*	6B	0.01	2.47	*Xgwm940b*	−0.37	0.08	B2
RPhr_S2	*QRPhr_S2.NIAB-6B*	6B	60.81	2.09	*Xbarc136*	8.31	0.28	O5
Phr_S2	*QPhr_S2.NIAB-7A*	7A	186.41	**3.14**	*Xgwm276*	−0.09	0.13	B2
PhrD_S2	*QPhrD_S2.NIAB-7B*	7B	106.81	2.06	*Xwmc182*	0.12	0.06	O5
RPhr_S2	*QRPhr_S2.NIAB-7B^xx^*	7B	150.61	2.31	*Xgwm46*	−4.71	0.08	B2

WueD_S2	*QWueD_S2.NIAB-2A^j^*	2A	166.21	2.33	*Xgwm1256*	−1.56	0.19	B2
WueD_S1	*QWueD_S1.NIAB-2B^k^*	2B	0.01	**3.71**	*Xbarc183*	−0.03	0.11	B2
WueD_S1	*QWueD_S1.NIAB-4B*	4B	31.71	2.6	*Xbarc199*	−0.02	0.08	B2
Wue_S1	*QWue_S1.NIAB-4B.1^aa^*	4B	36.51	**3.06**	*Xgwm710b*	−0.01	0.1	B2
Wue_S1	*QWue_S1.NIAB-4B.2^bb^*	4B	43.31	2.12	*Xgwm1167*	−0.01	0.07	B2
WueD_S2	*QWueD_S2.NIAB-5A*	5A	259.01	**4.5**	*Xwmc727*	1.88	0.22	O5
Wue_S2	*QWue_S2.NIAB-5A^ii^*	5A	269.61	2	*Xgwm126*	0.93	0.08	O5
Wue_S2	*QWue_S2.NIAB-5B^mm^*	5B	158.91	2.12	*Xgwm408*	−0.88	0.06	B2
Wue_S1	*QWue_S1.NIAB-7A*	7A	18.01	2.69	*Xwmc283*	−0.01	0.09	B2
WueD_S2	*QWueD_S2.NIAB-7B^xx^*	7B	149.61	2	*Xgwm46*	−0.86	0.06	B2
Wue_S1	*QWue_S1.NIAB-7B*	7B	329.61	2.08	*Xbarc32*	0.01	0.07	O5
**Spike Related Traits**								
Sl_S1	*QSl_S1.NIAB-2A^g^*	2A	76.41	**6.89**	*Xgwm122*	0.46	0.21	O5
SlD_S1	*QSlD_S1.NIAB-2B*	2B	33.21	**4.23**	*Xbarc55*	−0.57	0.16	B2
RSl_S1	*QRSl_S1.NIAB-3A^q^*	3A	61.11	2.1	*Xwmc50*	−5.64	0.13	B2
SlD_S2	*QSlD_S2.NIAB-4B^cc^*	4B	151.81	2.34	*Xgwm940a*	−0.18	0.08	B2
Sl_S1	*QSl_S1.NIAB-4B*	4B	154.81	**3.8**	*Xgwm935a*	−0.33	0.19	B2
Sl_S2	*QSl_S2.NIAB-5B*	5B	33.31	**3.04**	*Xgwm1180*	−0.24	0.1	B2
SlD_S2	*QSlD_S2.NIAB-7A^vv^*	7A	217.81	2.13	*Xgwm332*	−0.17	0.07	B2
Sl_S2	*QSl_S2.NIAB-7A*	7A	278.21	**4.05**	*Xgwm1061*	−0.27	0.14	B2
RSl_S2	*QRSl_S2.NIAB-7B^ww^*	7B	89.91	**3.86**	*Xgwm573*	3.11	0.18	O5
RSl_S1	*QRSl_S1.NIAB-7B*	7B	117.61	2.38	*Xgwm1184*	−5.05	0.1	B2

SwD_S1	*QSwD_S1.NIAB-1A^a^*	1A	27.11	2.71	*Xwmc312*	0.18	0.09	O5
Sw_S1	*QSw_S1.NIAB-1A*	1A	30.11	**5.25**	*Xwmc93a*	0.29	0.19	O5
SwD_S1	*QSwD_S1.NIAB-2A^g^*	2A	76.41	2.22	*Xgwm122*	0.14	0.06	O5
RSw_S1	*QRSw_S1.NIAB-2A^g^*	2A	76.41	**3.38**	*Xgwm122*	5.1	0.1	O5
SwD_S1	*QSwD_S1.NIAB-3A^q^*	3A	68.51	2.55	*Xwmc50*	−0.16	0.08	B2
RSw_S1	*QRSw_S1.NIAB-3A^q^*	3A	72.51	**3.02**	*Xwmc50*	−5.64	0.12	B2
Sw_S1	*QSw_S1.NIAB-3A^s^*	3A	97.81	2.62	*Xbarc356*	0.2	0.08	O5
SwD_S1	*QSwD_S1.NIAB-3B^u^*	3B	13.91	2.82	*Xgwm685*	0.18	0.08	O5
RSw_S1	*QRSw_S1.NIAB-3B^w^*	3B	194.81	2.28	*Xgwm299*	−5.65	0.12	B2
SwD_S1	*QSwD_S1.NIAB-5A^dd^*	5A	0.01	2.94	*Xgwm154*	0.18	0.09	O5
SwD_S2	*QSwD_S2.NIAB-5A^ii^*	5A	277.71	2.14	*Xgwm126*	−0.12	0.09	B2
RSw_S1	*QRSw_S1.NIAB-5A^jj^*	5A	337.41	**3.1**	*Xbarc261*	−5.74	0.13	B2
RSw_S2	*QRSw_S2.NIAB-5B.1^oo^*	5B	237.11	2.42	*Xgwm1072*	4.68	0.09	O5
RSw_S2	*QRSw_S2.NIAB-5B.2^pp^*	5B	260.01	**3.65**	*Xgwm790*	5.87	0.15	O5
SwD_S1	*QSwD_S1.NIAB-5B*	5B	270.91	2.23	*Xgwm605*	0.14	0.06	O5
SwD_S2	*QSwD_S2.NIAB-7A^uu^*	7A	57.81	2	*Xwmc405*	−0.12	0.08	B2
RSw_S1	*QRSw_S1.NIAB-7A*	7A	98.81	2.59	*Xgwm710a*	−4.37	0.07	B2
RSw_S1	*QRSw_S1.NIAB-7B^ww^*	7B	92.21	**3.09**	*Xgwm573*	5.36	0.09	O5

SpsD_S1	*QSpsD_S1.NIAB-1A^a^*	1A	25.11	2.41	*Xwmc312*	2.11	0.06	O5
RSps_S1	*QRSps_S1.NIAB-1B^c^*	1B	104.61	2.12	*Xwmc419*	−4.03	0.06	B2
Sps_S1	*QSps_S1.NIAB-1B.1^d^*	1B	143.81	2.5	*Xwmc156*	3.22	0.11	O5
Sps_S1	*QSps_S1.NIAB-1B.2*	1B	154.61	2.75	*Xwmc548*	2.91	0.08	O5
Sps_S2	*QSps_S2.NIAB-3A^p^*	3A	56.11	2.5	*Xbarc45*	2.63	0.1	O5
SpsD_S1	*QSpsD_S1.NIAB-3A^q^*	3A	68.51	**3.6**	*Xwmc50*	−2.6	0.1	B2
Sps_S1	*QSps_S1.NIAB-3B*	3B	8.31	2	*Xgwm625*	2.4	0.06	O5
SpsD_S1	*QSpsD_S1.NIAB-3B^u^*	3B	13.91	**3.61**	*Xgwm685*	2.67	0.1	O5
RSps_S1	*QRSps_S1.NIAB-3B^w^*	3B	182.81	2.63	*Xgwm299*	4.51	0.08	O5
Sps_S2	*QSps_S2.NIAB-4B*	4B	155.91	2.94	*Xwmc428*	−3.29	0.09	B2
SpsD_S1	*QSpsD_S1.NIAB-5A^dd^*	5A	1.01	2.84	*Xgwm154*	2.43	0.08	O5
Sps_S1	*QSps_S1.NIAB-5B.1^kk^*	5B	13.41	2.3	*Xgwm540a*	−3.34	0.07	B2
RSps_S1	*QRSps_S1.NIAB-5B^ll^*	5B	132.21	**4.41**	*Xgwm1043*	8.72	0.33	O5
RSps_S2	*QRSps_S2.NIAB-5B.2^ll^*	5B	143.21	2.14	*Xgwm1043*	3.9	0.07	O5
SpsD_S2	*QSpsD_S2.NIAB-5B*	5B	218.31	**3.22**	*Xbarc232*	2	0.11	O5
RSps_S2	*QRSps_S2.NIAB-5B.2^oo^*	5B	232.71	2.83	*Xgwm1072*	4.41	0.09	O5
Sps_S2	*QSps_S2.NIAB-5B^oo^*	5B	233.11	2.27	*Xgwm1072*	−2.17	0.07	B2
Sps_S1	*QSps_S1.NIAB-5B.2*	5B	273.91	2.85	*Xbarc243*	2.9	0.09	O5
SpsD_S2	*QSpsD_S2.NIAB-7B^yy^*	7B	172.61	2.1	*Xwmc218*	−2.36	0.07	B2

RTkw_S1	*QRTkw_S1.NIAB-1B^c^*	1B	102.61	2.05	*Xwmc419*	−2.01	0.08	B2
TkwD_S1	*QTkwD_S1.NIAB-1B.1*	1B	221.11	2.26	*Xgwm268*	1.56	0.09	O5
TkwD_S1	*QTkwD_S1.NIAB-1B.2*	1B	230.31	2.21	*Xgwm153*	1.32	0.06	O5
Tkw_S1	*QTkw_S1.NIAB-2A^f^*	2A	17.51	2.11	*Xgwm726*	1.88	0.08	O5
RTkw_S2	*QRTkw_S2.NIAB-4B.1^aa^*	4B	35.71	**3.98**	*Xgwm710b*	−1.93	0.11	B2
Tkw_S2	*QTkw_S2.NIAB-4B^aa^*	4B	35.71	2.74	*Xgwm710b*	1.98	0.09	O5
RTkw_S2	*QRTkw_S2.NIAB-4B.2^bb^*	4B	42.51	**4.24**	*Xgwm1167*	−1.98	0.12	B2
RTkw_S1	*QRTkw_S1.NIAB-5A^ff^*	5A	124.71	**3.22**	*Xgwm1236*	−2.51	0.12	B2
TkwD_S1	*QTkwD_S1.NIAB-5A^ff^*	5A	126.71	2	*Xgwm1236*	−1.38	0.07	B2
TkwD_S2	*QTkwD_S2.NIAB-5A^ii^*	5A	277.71	2.62	*Xgwm126*	−1.85	0.1	B2
RTkw_S2	*QRTkw_S2.NIAB-5B*	5B	62.01	2.01	*Xbarc128*	−1.38	0.05	B2
Tkw_S1	*QTkw_S1.NIAB-6B^rr^*	6B	144.11	2.55	*Xbarc178*	1.86	0.08	O5
TkwD_S1	*QTkwD_S1.NIAB-6B^ss^*	6B	160.61	2.74	*Xgwm889*	1.61	0.09	O5
RTkw_S2	*QRTkw_S2.NIAB-7A*	7A	113.41	2.14	*Xwmc603*	−1.35	0.05	B2
Tkw_S1	*QTkw_S1.NIAB-7A^vv^*	7A	220.81	2.93	*Xgwm332*	2.03	0.1	O5
RTkw_S1	*QRTkw_S1.NIAB-7B*	7B	275.21	2.66	*Xbarc258*	−2.09	0.09	B2
**Yield and Related Traits**								
RBm_S1	*QRBm_S1.NIAB-3A^o^*	3A	16.61	2.32	*Xgwm757*	−1.29	0.06	B2
RBm_S1	*QRBm_S1.NIAB-3B^t^*	3B	11.91	2.06	*Xgwm285*	−1.22	0.05	B2
RBm_S2	*QRBm_S2.NIAB-3B^v^*	3B	88.11	2.85	*Xbarc344*	3.88	0.1	O5
Bm_S2	*QBm_S2.NIAB-3B^v^*	3B	89.11	2.22	*Xbarc344*	−349.5	0.07	B2
BmD_S1	*QBmD_S1.NIAB-4B*	4B	88.31	**3.56**	*Xgwm998*	130.01	0.11	O5
RBm_S1	*QRBm_S1.NIAB-5A^ee^*	5A	46.91	2.4	*Xgwm129*	−1.39	0.07	B2
Bm_S2	*QBm_S2.NIAB-5B^oo^*	5B	232.71	2.65	*Xgwm1072*	−389.16	0.08	B2
RBm_S2	*QRBm_S2.NIAB-5B^oo^*	5B	232.71	2.03	*Xgwm1072*	3.04	0.07	O5
RBm_S1	*QRBm_S1.NIAB-5B^pp^*	5B	259.01	**3.45**	*Xgwm790*	−1.78	0.11	B2
RBm_S1	*QRBm_S1.NIAB-7A*	7A	213.91	2.83	*Xbarc253*	−1.46	0.08	B2
RBm_S2	*QRBm_S2.NIAB-7A*	7A	91.51	2.21	*Xbarc1025*	−3.38	0.08	B2
BmD_S2	*QBmD_S2.NIAB-7A^uu^*	7A	52.81	2.73	*Xwmc405*	−150.39	0.13	B2
Bm_S1	*QBm_S1.NIAB-7B.1*	7B	137.51	**3.06**	*Xgwm540b*	−558.27	0.1	B2
Bm_S1	*QBm_S1.NIAB-7B.2*	7B	164.41	**3.16**	*Xwmc476*	640.83	0.13	O5
RBm_S2	*QRBm_S2.NIAB-7B.1^yy^*	7B	172.61	**3.55**	*Xwmc218*	−4.24	0.12	B2
RBm_S2	*QRBm_S2.NIAB-7B.2*	7B	179.41	**3.39**	*Xgwm963b*	−4.14	0.11	B2
RBm_S2	*QRBm_S2.NIAB-7B.3^zz^*	7B	184.41	2.43	*Xgwm1085*	−3.75	0.08	B2
Bm_S1	*QBm_S1.NIAB-7B.3^aaa^*	7B	218.21	2	*Xbarc276*	−334.54	0.07	B2
BmD_S2	*QBmD_S2.NIAB-7B^aaa^*	7B	221.11	2	*Xbarc276*	−113.28	0.07	B2

RYd_S2	*QRYd_S2.NIAB-1B^b^*	1B	95.61	2.07	*Xgwm784*	3.61	0.06	O5
YdD_S1	*QYdD_S1.NIAB-2A^f^*	2A	26.01	2.65	*Xgwm726*	37.77	0.09	O5
YdD_S2	*QYdD_S2.NIAB-2A^i^*	2A	124.61	2.51	*Xgwm895*	34.89	0.08	O5
RYd_S1	*QRYd_S1.NIAB-2A^i^*	2A	125.21	2.44	*Xgwm895*	−4.07	0.07	B2
Yd_S2	*QYd_S2.NIAB-3A^p^*	3A	52.11	2.38	*Xbarc45*	96.11	0.07	O5
RYd_S2	*QRYd_S2.NIAB-3B^x^*	3B	205.71	**3.58**	*Xgwm547*	−4.91	0.11	B2
Yd_S2	*QYd_S2.NIAB-3B^x^*	3B	205.71	2.2	*Xgwm547*	91.54	0.06	O5
Yd_S1	*QYd_S1.NIAB-5A^gg^*	5A	158.81	2.09	*Xbarc165*	99.37	0.07	O5
RYd_S2	*QRYd_S2.NIAB-5A^hh^*	5A	193.01	**4.04**	*Xgwm1171b*	−7.68	0.3	B2
YdD_S1	*QYdD_S1.NIAB-5B^ll^*	5B	143.21	2.66	*Xgwm1043*	38.45	0.09	O5
YdD_S1	*QYdD_S1.NIAB-6B*	6B	9.91	2.29	*Xgwm1199*	−35.55	0.08	B2
RYd_S1	*QRYd_S1.NIAB-6B*	6B	121.71	2.09	*Xgwm963a*	−3.85	0.07	B2
Yd_S1	*QYd_S1.NIAB-7B^aaa^*	7B	218.21	2.52	*Xbarc276*	−117.78	0.08	B2
YdD_S2	*QYdD_S2.NIAB-7B^aaa^*	7B	219.11	**3**	*Xbarc276*	−39.23	0.09	B2

RHI_S2	*QRHI_S2.NIAB-1B^b^*	1B	93.11	2.22	*Xgwm784*	9.7	0.07	O5
RHI_S2	*QRHI_S2.NIAB-2B^n^*	2B	108.81	2.4	*Xgwm47*	−10.1	0.08	B2
RHI_S2	*QRHI_S2.NIAB-3B.1^v^*	3B	89.81	2.91	*Xbarc344*	−10.71	0.09	B2
HI_S2	*QHI_S2.NIAB-3B^x^*	3B	205.71	2.42	*Xgwm547*	2.03	0.08	O5
RHI_S2	*QRHI_S2.NIAB-3B.2^x^*	3B	205.71	2.8	*Xgwm547*	−10.17	0.09	B2
HID_S2	*QHID_S2.NIAB-4B*	4B	130.61	2.5	*Xbarc60*	2.02	0.08	O5
HID_S1	*QHID_S1.NIAB-4B^cc^*	4B	151.21	2.59	*Xgwm940a*	3.31	0.09	O5
HI_S1	*QHI_S1.NIAB-5B*	5B	228.61	2.77	*Xwmc28*	3.19	0.1	O5
HID_S2	*QHID_S2.NIAB-5B^ll^*	5B	143.21	**3.3**	*Xgwm1043*	2.24	0.11	O5
HID_S2	*QHID_S2.NIAB-7A*	7A	108.61	2.42	*Xbarc108*	1.8	0.07	O5
RHI_S2	*QRHI_S2.NIAB-7B^zz^*	7B	184.41	2.91	*Xgwm1085*	11.25	0.09	O5
HI_S1	*QHI_S1.NIAB-7B*	7B	336.01	2.42	*Xwmc273*	2.95	0.08	O5
**Stress Indices**								
STI_S2	*QSTI_S2.NIAB-4A.1*	4A	44.51	2.73	*Xbarc246*	0.06	0.09	O5
STI_S2	*QSTI_S2.NIAB-4A.2*	4A	100.81	2.13	*Xgwm1234*	−0.06	0.06	B2
STI_S1	*QSIT_S1.NIAB-6B^ss^*	6B	159.91	2.14	*Xgwm889*	0.06	0.07	O5
STI_S2	*QSTI_S2.NIAB-7A^tt^*	7A	0.01	**3.36**	*Xgwm1171a*	−0.07	0.11	B2

MP_S1	*QMP_S1.NIAB-1B*	1B	72.01	2.28	*Xgwm762*	87.36	0.08	O5
MP_S2	*QMP_S2.NIAB-1B*	1B	116.81	2.21	*Xbarc207*	−61.49	0.08	B2
MP_S2	*QMP_S2.NIAB-2A^j^*	2A	158.21	2.04	*Xgwm1256*	97.14	0.22	O5
MP_S2	*QMP_S2.NIAB-3B.1^w^*	3B	194.81	2.87	*Xgwm299*	82.74	0.16	O5
MP_S2	*QMP_S2.NIAB-3B.2^x^*	3B	205.71	**3.72**	*Xgwm547*	75.53	0.13	O5
MP_S1	*QMP_S1.NIAB-5A^gg^*	5A	158.81	2.4	*Xbarc165*	63.78	0.08	O5

ST_S2	*QST_S2.NIAB-3B^w^*	3B	192.81	2.22	*Xgwm299*	99.6	0.11	O5
ST_S1	*QST_S1.NIAB-5A^gg^*	5A	165.81	2.12	*Xbarc165*	112.92	0.11	O5
ST_S2	*QST_S2.NIAB-5A^hh^*	5A	179.81	**3.43**	*Xgwm1171b*	116.02	0.13	O5

DRI_S2	*QDRI_S2.NIAB-2B^m^*	2B	52.21	2.02	*Xgwm972*	59.7	0.06	O5
DRI_S2	*QDRI_S2.NIAB-4A*	4A	64.31	2.47	*Xbarc343*	82.63	0.1	O5
DRI_S2	*QDRI_S2.NIAB-5A^hh^*	5A	186.01	2.59	*Xgwm1171b*	−81.06	0.11	B2
DRI_S1	*QDRI_S1.NIAB-5A^hh^*	5A	188.01	**4.63**	*Xgwm1171b*	−111.01	0.26	B2
DRI_S1	*QDRI_S1.NIAB-5B^oo^*	5B	232.71	2.44	*Xgwm1072*	63.65	0.08	O5

**Table 3 ijms-21-02372-t003:** Percentage increase or decrease in yield under various conditions with respect to *Xgwm895* at 2A and *Xbarc276* at 7B.

Trait	*Xgwm895,* 2A		*Xbarc276,* 7B		Combination		
	A allele	B allele	A allele	B allele	AA	BB	AB
Yd_S1	5.5	−9.31	5.5	6.6	−4	−1.89	2.16
YdD_S1	0.18	−1.6	0.18	0.66	0.61	−1.83	−0.15
RYd_S1	−6.65	6.82	−6.65	−4.66	1.56	2.26	−14.1
Yd_S2	6.16	−9.35	6.16	2.57	0.92	−10.78	10.2
YdD_S2	5.27	−8.4	5.27	4.39	22.98	−10.43	11.3
RYd_S2	−1.59	1.85	−1.59	2.61	−3.04	4.74	−0.39

**Table 4 ijms-21-02372-t004:** Absolute and relative traits measured for RIL population in 2016–17 (trait abbreviation followed by _S1) and 2017–18 (trait abbreviation followed by _S2) in control and drought and their time of measurement each season.

	Traits	Control	Drought	Relative	Time
Agronomic Traits					
1	Germination	Gr_S1, Gr_S2	GrD_S1, GrD_S2	RGr_S1, RGr_S2	24th Nov
2	Heading date	Hd_S1, Hd_S2	HdD_S1, HdD_S2	RHd_S1, RHd_S2	50% spike emerg
3	Plant height	Ph_S1, Ph_S2	PhD_S1, PhD_S2	RPh_S1, RPh_S2	Before harvest
Physiological Traits					
4	Transpiration rate	Tr_S1, Tr_S2	TrD_S1, TrD_S2	RTr_S1, RTr_S1	7^th^ Feb
5	Stomatal conductance	Sc_S1, Sc_S2	ScD_S1, ScD_S2	RSc_S1, RSc_S2	7^th^ Feb
6	Photosynthetic rate	Phr_S1, Phr_S2	PhrD_S1, PhrD_S2	RPhr_S1, RPhr_S2	7^th^ Feb
7	Water use efficiency	Wue_S1, Wue_S2	WueD_S1, WueD_S2	RWue_S1, RWue_S2	7^th^ Feb
Spike-Related Traits					
8	Spike length	Sl_S1, Sl_S2	SlD_S1, SlD_S2	RSl_S1, RSl_S2	After harvest
9	Spike weight	Sw_S1, Sw_S2	SwD_S1, SwD_S2	RSw_S1, RSw_S2	After harvest
10	Seeds per spike	Sps_S1, Sps_S2	SpsD_S1, SpsD_S2	RSps_S1, RSps_S2	After harvest
11	Thousand kernel weight	Tkw_S1, Tkw_S2	TkwD_S1, TkwD_S2	RTkw_S1, RTkw_S2	After harvest
	**Traits**	**Control**	**Drought**	**Time**	
Yield and Related Traits					
12	Biomass	Bm_S1, Bm_S2	BmD_S1, BmD_S2	After harvest	
13	Yield	Yd_S1, Yd_S2	YdD_S1, YdD_S2	After harvest	
14	Harvest Index	HI_S1, HI_S2	HID_S1, HID_S2	After harvest	
	**Index**	**Abbrev.**	**Formula**	**Time**	
Stress Indices					
15	Stress Tolerance Index (STI)	STI_S1, STI_S2	(Yp×Ys)(Yp¯)2	After harvest	
16	Mean Productivity (MP)	MP_S1, MP_S2	(Yp+Ys)2	After harvest	
17	Stress Tolerance (ST)	ST_S1, ST_S2	Yp−Ys	After harvest	
18	Stress Susceptibility Index (SSI)	SSI_S1, SSI_S2	[1−(YsYp)]/[1−(Ys¯Yp¯)]	After harvest	
19	Drought Resistance Index (DRI)	DRI_S1, DR_S2	Ys×(YsYp)/Ys¯	After harvest	
where *Ys* = yield under stress, $\bar{Ys}$ = mean yield under stress, *Yp* = yield under control and $\bar{Yp}$ = mean yield under control					

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
