# Peer review of "Mapping of QTLs Associated with Yield and Yield Related Traits in Durum Wheat (*Triticum durum* Desf.) Under Irrigated and Drought Conditions"

_ijms, 2020, doi:10.3390/ijms21072372_

Round 1

Reviewer 1 Report

The manuscript is well written and covers properly described topic.

Drought caused by soil water deficit is one of the most important constraints for crop improvement all over the world. Several research groups put a lot of efforts towards research focused on the search for traits that lead to better yield stability under water deficit conditions. Large-scale datasets such as presented in the manuscript is an excellent tool for other research groups, but what is most important it delivers valuable information on QTLs related to environmental stresses. It is very important to understand how plants respond to environmental signals to regulate their development and immunity and to maintain their optimal growth, reproduction, and fitness. Within presented study, a large amount of information was collected for durum wheat in two seasons. It is impressive how big dataset authors have generated by analysing a population of 114 RILs and performing 265 assays. The most essential finding of this work are the identified 221 QTLs. This information on the components of yield and agronomic traits under irrigated and drought conditions provides new sights into the genetic architecture of yield and related traits in durum wheat. What is more important, it can be further utilised in breeding programmes, not only of durum wheat, but several other cereals. Such information is valuable in order to better prepare us for global climate changes. Results can be incorporated in crop improvement programs and crop breeders will have the potential to improve their efficiency of selection based on analysed traits. Authors, through their work, deliver important information and tools. The experimental design was adequate, and the number of analysed parameters covered all aspects of the research problem. Also, statistical analysis and the overall data analysis was done correctly. Essential for such studies is the repetition of the experiments in two seasons, which was also performed.

The manuscript should be carefully reviewed for English and grammatic correctness.

Author Response

Thank you very much for your encouraging remarks regarding the MS. We appreciate your efforts in reviewing the MS. We have made needful changes in the light of your comments.

Reviewer 2 Report

The main aim of the paper is map QTL associated with yield in durum wheat.

The information is very complete and can be useful about these markers and the traits that they are related. However, more discussion is needed, and the conclusions need to be expanded and more comparisons with other studies and results are needed. Information should be given more precisely and easier for the reader.

Specific comments

Line 38. Reference needed of pasta purchased.

Line 74. In the last years, because at the beginning high improvements in the HI were made.

Line 129. Abbreviatures explanation needed. Each table needs to be auto explicative.

Line 136. If germination was lower that 100% it means that some RILs were not tested? were replaced? The same in both seasons?

Line 145. Statistical differences between seasons? Information of the table is mainly repeated in the text. Rewrite to avoid it.

Line 133 and ahead. Comparisons are made between season 1 and 2. But where is the comparison between control and drought data? That will be much more interesting!

Line 219. Again, information of the text and the table is redundant.

Line 459. References were drought tolerance of Omrabi5 is described?

Line 470. “one replicate in control and three under drought” why only one control? How many plants per RILs? It means 1 plant per RIL in control and three under drought’ Explain, please. Control and drough experiments were in the same field? Precipitation data per season? Distribution and quantity of rain per season?

Line 503. Biomass included the roots?

Author Response

Thank you very much for your precious comments. We have tackled each of the comment as per the suggestion of the reviewer-2. Overall, objective of this study has been clarified by improving the introduction part as well as more explanation of experimental setup has been added. Results have been improved for understanding of the readers and discussion part has been improved extensively keep in view the results. Furthermore, point to point reply of each comment is as follow;

Specific comments

  1. Line 38. Reference needed of pasta purchased.

We have rewritten the information provided at line 38 as the word purchased needed to be replaced with produced. This line and the following line are from the same source which is referred by reference # 5. Hence the new sentence is as follows:

A total of 9.3 and 10.5 million tons pasta was produced in 2001 and 2003, respectively. By 2013, its production reached to 13.5 million tons [5] providing an insight into the growing global durum wheat.

  1. Line 74. In the last years, because at the beginning high improvements in the HI were made.

Yes, the reviewer has pointed out the shortcoming in this case. We have re-written the sentence.

After 1970s, the attained genetic gains in durum grain Yd are probably attributed to well-adjusted progress in fertility through distribution of relatively higher assimilates to the ear and the growing tillers which increased total biomass whereas HI remained the same [19] probably because ratio of improvement of grain yield to biological yield remained same.

  1. Line 129. Abbreviatures explanation needed. Each table needs to be auto explicative.

The following has been incorporated in Tables

Gr = germination %, Tr = Transpiration rate, Sc = Stomatal conductance, Phr = photosynthetic rate, Wue = water use efficiency, Hd = heading time, Ph = plant height, Sl = spike length, Sw = Spike weight, Sps = seeds per spike, Bm = Biomass, Yd = Yield, HI = harvest index, Tkw = Thousand kernal weight, STI = stress tolerance index, MP = mean productivity, ST =stress tolerance, SSI = stress susceptibility index, DRI = drought resistance index and P values were also included in the same table along with respective means.

  1. Line 136. If germination was lower that 100% it means that some RILs were not tested? were replaced? The same in both seasons?

Our threshold criterion was at least 80% germination in control. There were 38 RILs in S1 (where majority ranged between 70-75% and only 4 RILs with 50-65% germination) and none in S2 where germination was < 80%; they were replaced with new seedlings grown in lab and transferred in the field to bring the germination up to 80%. Correspondingly, their drought counterparts were also replaced accordingly to avoid any biasness in other data. Furthermore, from a total of 114 RILs, three RILs were completely missed (RIL # 13, 80 and 111).      

  1. Line 145. Statistical differences between seasons? Information of the table is mainly repeated in the text.

To avoid repetition, we have combined Tables 1 and 2 and made it one single Table to compare the phenotypes as well as their differences from genotype, season, treatment and their interaction perspectives. Significant differences were explained concisely to minimize the repetition.

  1. Line 133 and ahead. Comparisons are made between season 1 and 2. But where is the comparison between control and drought data? That will be much more interesting!

Thank you very much for this comment. Needful is done in the new edited MS

  1. Line 219. Again, information of the text and the table is redundant.

The paragraphs after line 219 till 259 explain QTLs for each trait a little. We just put only the mandatory information here for the readers’ assistance so that when we are discussing the results, readers might not feel the need to refer to Table again and again. Moreover, this explanation makes the MS more informative as here we have discussed the QTLs for group of related traits. In addition, some readers like to read in text and some readers like to only glance at the tables. Here, we have tried to cater to the needs of all kinds of readers. 

  1. Line 459. References were drought tolerance of Omrabi5 is described?

The reference for both Omrabi-5 and Belikh 2 is [17] Nachit, M.M.; Elouafi, I. Durum wheat adaptation in the Mediterranean dryland: Breeding, stress physiology, and molecular markers. In Challenges and Strategies of Dryland Agriculture, (challengesandst) 2004, pp.203-218 which is added in text

  1. Line 470. “one replicate in control and three under drought” why only one control? How many plants per RILs? It means 1 plant per RIL in control and three under drought’ Explain, please. Control and drough experiments were in the same field? Precipitation data per season? Distribution and quantity of rain per season?

There was one replicate in control in three in drought because of the limitations of field as well as labor to maintain the experiment and data collection. Moreover, we did not measure the yield on per plant basis, rather we measured it on plot basis where each plot carried 20 plants per line (2 lines per RIL/replicate = 40 plants). The drought and control experiments were in the same field where the control experiment was separated from the drought by making a 2 ft high ridge partition and there was a gap of at least 5 meters between drought and control treatments. Furthermore, there was also a gap of 5 meters between drought treatment and water channel to avoid any leach down effect that might have an effect on drought treatments. To add to it, we planted brassica in the leftover area so that the leached water from the water channel and control experiment is used up before it reached to the drought treatment. Below is the layout plan of our experiments which was followed in the exact same way for both seasons. The total rainfall in S1 was 36.25 distributed over the period of 14 days (from Nov 2016-May 2017) where the maximum rain was received in April-17 (16.58 mm). In S2, total rainfall was 42.48 distributed over the period of 30 days with maximum rainfall received in April-18 (12.21 mm).

Rainfall data of the two growing seasons (S1 and S2) of Faisalabad

2016-17 (S1)

2017-18 (S2)

Amount in mm

Days

Amount in mm

Days

Nov

0

0

0.44

0

Dec

0

0

0.06

0

Jan

11.14

5

0

0

Feb

1.22

1

7.25

3

Mar

4.71

2

10.56

7

April

16.58

4

12.21

15

May

2.6

2

11.96

5

Total

36.25

14

42.48

30

Temperature data in °C of the two growing seasons (S1 and S2) of Faisalabad

2016-17 (S1)

2017-18 (S2)

Min.

Max.

Mean

Min.

Max.

Mean

Nov

16

31

22

17

19

22

Dec

12

27

18

13

24

17

Jan

9

21

13

10

25

15

Feb

11

27

18

13

26

18

March

14

31

22

17

32

24

April

23

39

31

23

36

29

May

29

42

36

30

41

36

Total

16.28

31.14

22.85

17.57

29

23

Lay out plan of the experiments.

  1. Line 503. Biomass included the roots?

Since we cut the complete rows with hand held sickle, it was not possible to take roots in to account. Hence, biomass only includes the above ground plant parts. The same has been added in the MS

Round 2

Reviewer 2 Report

Comments have been included or corrected